# Distribution of the four type VI secretion systems in *Pseudomonas aeruginosa* and classification of their core and accessory effectors

Antonia Habich[1,2], Verónica Chaves Vargas [1,2], Luca A. Robinson [3], Luke P. Allsopp [3] & Daniel Unterweger [1,2] ✉

Bacterial type VI secretion systems (T6SSs) are puncturing molecular machines that transport effector proteins to kill microbes, manipulate eukaryotic cells, or facilitate nutrient uptake. How and why T6SS machines and effectors differ within a species is not fully understood. Here, we applied molecular population genetics to the T6SSs in a global population of the opportunistic pathogen *Pseudomonas aeruginosa*. We reveal varying occurrence of up to four distinct T6SS machines. Moreover, we define conserved core T6SS effectors, likely critical for the biology of *P. aeruginosa*, and accessory effectors that can exhibit mutual exclusivity between strains. By ancestral reconstruction, we observed dynamic changes in the gain and loss of effector genes in the species' evolutionary history. Our work highlights the potential importance of T6SS intraspecific diversity in bacterial ecology and evolution.

In natural environments with competing organisms and low nutrient supply, bacteria survive by killing others. One sophisticated molecular machine to do so is the type VI secretion system (T6SS)[1,2]. It resembles the structure of an inverted phage tail and enables the secretion of effector proteins directly into other cells[3,4] or extracellularly[5]. The T6SS are present in Gram-negative bacteria across phyla and attributed to a variety of bacterial lifestyles, from symbiosis of host-associated bacteria to pathogen virulence[4,6–8]. Little is known about the distribution of this molecular machine and its effectors across individual bacterial species.

Despite the high conservation of the T6SS apparatus, its effector proteins exhibit remarkable diversity. Multiple modes of effector coupling to the T6SS structure enable effectors to either be secreted within the tube or at various positions at the tip of the secretion system[4,9–11]. Effectors with a variety of molecular activities have been reported, including lipases, nucleases, muramidases, inhibition of translation, and pore formation[12–17]. Depending on which effectors are secreted, and where they are delivered, effectors can kill bacteria[3] or fungi[18], manipulate eukaryotic cells[4], and aid nutrient acquisition[19].

How a bacterium benefits from T6SS-mediated protein secretion in its environment, therefore, depends on its effectors.

*Pseudomonas aeruginosa* possesses some of the best-studied T6SSs in the field. However, our knowledge is dominated by two reference strains (PAO1 and PA14), both of which have 3 distinct T6SSs machines (referred to as H1-, H2-, and H3-T6SS) and secrete a repertoire of 22 and 24 currently defined effectors, respectively (Supplementary Data 1). Several studies show that the less studied *P. aeruginosa* isolates differ in their T6SS effectors from the two reference strains. For example, the gene encoding the effector PldA was observed to be present in some and absent from others[20–22]. When characterising the effectors Tse6 and Tse7 biochemically, Ahmad et al. and Pissaridou and Allsopp et al. noted that strains differed in the genes encoding for these effectors[23,24]. Moreover, Robinson et al. and Habich et al. recently reported on the diversity of T6SS effectors in a local population of clinical isolates[25,26]. Despite these individual reports, how and why T6SSs apparatus and effector occurrence differs across the species remains unclear.

[1]Institute for Experimental Medicine, Kiel University, Kiel, Germany. [2]Max Planck Institute for Evolutionary Biology, Plön, Germany. [3]National Heart and Lung Institute, Imperial College London, London, UK. ✉e-mail: d.unterweger@iem.uni-kiel.de

In this study, we applied molecular population genetics to a dataset of ~2000 phylogenetically diverse *P. aeruginosa* strains (Supplementary Data 2–4), focusing specifically on their T6SS-encoding genes. The publicly available genomes of the dataset are of high quality and represent the global *P. aeruginosa* population with strains from every continent, from humans through to animals and the environment, and exhibit remarkable phylogenetic diversity. First, we report on a fourth T6SS apparatus gene cluster in *P. aeruginosa* and analyse the taxonomic distribution of all four apparatus gene clusters – revealing unprecedented intraspecific variation in their occurrence. Second, we identify the T6SS effector repertoire of a global *P. aeruginosa* population and introduce the power of core and accessory effectors in the research field of the type VI secretion system. Third, we reconstruct the evolutionary history of effector genes within the *P. aeruginosa* species by determining the effectors of the most recent common ancestor (MRCA) and measuring dynamic gene gain and loss during diversification of the species. Fourth, we present the intraspecific diversity of T6SS effector sets arising from the presence and absence of individual effector genes across the genome and reveal the most common combination of effectors amongst *P. aeruginosa* strains. Fifth, we identify an effector with altered occurrence in the genomes obtained from strains isolated from human, animal, and environmental

origin. This work includes one of the largest pangenome analyses of *P. aeruginosa* to date and advances our understanding of the intraspecific distribution of genes encoding diverse T6SS apparatus and effector proteins that range in their function from nutrient acquisition to bacterial killing.

## Results

### Four type VI secretion system apparatus gene clusters are present in *P. aeruginosa*

Since its discovery in 2006, it has been established dogma that there are three distinct T6SS apparatus gene clusters in *P. aeruginosa*[2] (Fig. 1a, b). The proteins encoded in these genes form three T6SS apparatuses, which are very similar in their macromolecular structure but differ in their regulation and the effector proteins secreted by each system (Supplementary Data 1, 5). Here, we identify for the first time a gene cluster of a fourth T6SS in a few strains of the species that is so far uncharacterised (Fig. 1c). This cluster includes genes for 12 of the 13 required structural T6SS components (one of them being *tssM4* recently mentioned in the strain LYSZa7 by Ren et al.[27]). The 13th (*hcp4*) is encoded just upstream and thus has all the components required for a functional system. The synteny is distinct from the three known apparatus gene clusters in this species, which also differ from each

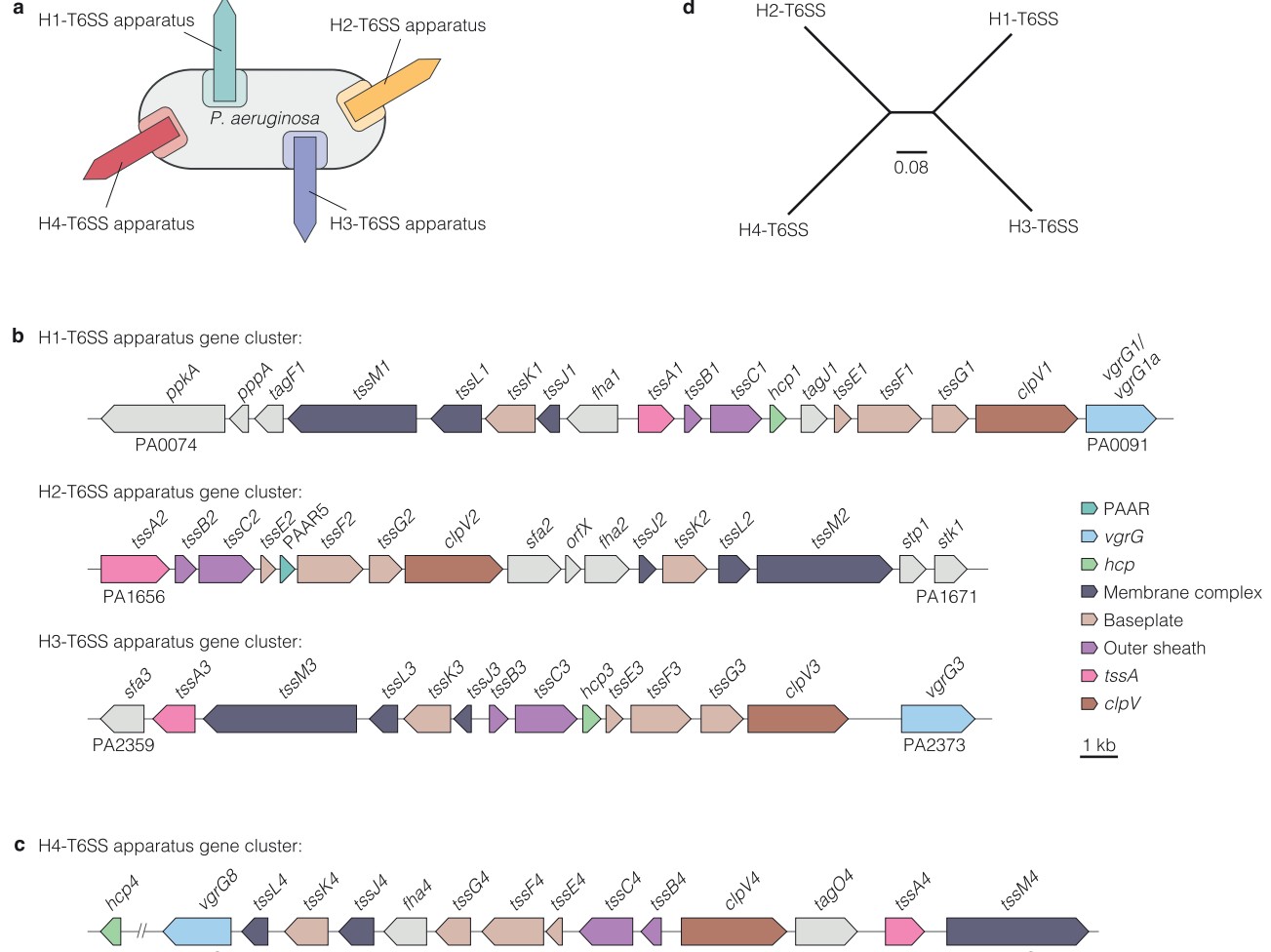

**Fig. 1 | Four T6SS apparatus gene clusters are present in the genome of *P. aeruginosa*. a** Schematic of a *P. aeruginosa* bacterium with four T6SS apparatuses. **b** Graphical depiction of the genomic organisation of the H1-, H2-, and H3-T6SS apparatus gene clusters from *P. aeruginosa* reference strain PAO1 showing distinct synteny. In each cluster, genes essential for protein assembly of a T6SS apparatus were coloured. **c** Graphical depiction of the H4-T6SS apparatus gene cluster of *P.* *aeruginosa* 60.1. **d** Maximum-likelihood phylogenetic tree based on the concatenated alignments of 12 genes (*tssA, tssB, tssC, tssE, tssF, tssG, tssJ, tssK, tssL, tssM, clpV,* and *hcp*) encoding for structural proteins of the four T6SSs of strain 60.1. The tree was built using the GTR + F + I model and is unrooted. Distances are shown in substitutions per site.

other in their order. Following the convention for *P. aeruginosa*, we label the gene cluster H4-T6SS. To probe the phylogenetic distinction between the four clusters, we built a tree based on the concatenated alignments of 12 apparatus genes (*tssA, tssB, tssC, tssE, tssF, tssG, tssJ, tssK, tssL, tssM, clpV,* and *hcp*) of each of the four clusters from strain 60.1. We found the H4-T6SS cluster in a separate branch from the H1-, H2-, and H3-T6SSs, showing that it is distinct and does not have recent common ancestry with the other three (Fig. 1d). Further confirming this result, phylogenetic trees of two apparatus genes *tssB* and *tssM* that encode proteins with essential T6SS functions as components of the outer sheath (TssB) and membrane complex (TssM) show clear cladding into four branches - one for each of the H1- to H4-T6SS (Supplementary Fig. 1). Analysis of the H4-T6SS with the SecReT6 database locates TssB4 of the H4-T6SS in the subgroup i1, making it most related to the H2-T6SS but clearly distinct in its organisation (Fig. 1b, c and Supplementary Fig. 2).

### H1-, H2-, H3-, and H4-T6SSs differ in their intraspecific distribution

To test for the distribution of the T6SS apparatus gene clusters within the species, we analysed their occurrence across strains. More specifically, we analysed a dataset of 1960 publicly available, high-quality *P. aeruginosa* genome sequences from strains of all continents and diverse sources (Supplementary Data 2–4). The H1-, H2-, or H3-T6SS apparatus gene clusters were found in at least 98% ($n = 1922/1960$) of the genomes, with the H2-T6SS having the highest occurrence (Fig. 2a, b and Supplementary Data 6). Remarkably, genes of the H4-T6SS were found in only 1% ($n = 19/1960$) of the strains analysed (Fig. 2a, b). Apparatus gene clusters of all four T6SSs were found in all phylogroups A, B, C1, and C2 (Fig. 2a). These findings are recapitulated when analysing the complete genomes of the dataset only ($n = 239$), excluding an artefact introduced by the genomes' assembly state (Supplementary Fig. 3).

The apparatus gene cluster of one of the H1-, H2-, or H3-T6SSs was found to be lacking in 2.4% ($n = 48/1960$) of the genomes analysed under study, which has not yet been described for this species (Fig. 2c). One example of a strain without an H1-T6SS is PAAK095 (accession number: GCF_013341315.1), a clinical isolate from a person with cystic fibrosis[28] (Fig. 2d). Examples for strains without the H2-T6SS or H3-T6SS are strains 4114358565 (GCF_004371095.1) and 836 (GCF_002194135.1), respectively. The three most used laboratory reference strains, PAO1, PA14, and PAK have all three T6SS apparatus gene clusters. No strain in this dataset was found with less than two T6SSs (Fig. 2c). The apparatus gene cluster of the H4-T6SS was only found in strains that additionally have the H1-, H2-, and H3-T6SS apparatus genes (Fig. 2a, d). In summary, we have found that the H2-T6SS was the most prevalent cluster, closely followed by the H1- and H3-T6SS and revealed a rare H4-T6SS occurrence in the global *P. aeruginosa* population.

### Multiple independent gain and loss events of apparatus gene clusters during species diversification

To investigate the source of diversity in the intraspecific occurrence of apparatus gene clusters, we used stochastic mapping. This method aims to reconstruct the evolutionary history of a gene on a phylogenetic tree of the species by determining the likelihood of the gene's presence at each node of the tree. The stochastic mapping showed the presence of the H1-, H2-, and H3-T6SS apparatus gene clusters in the MRCA of *P. aeruginosa* and descendant strains (Supplementary Fig. 4 and Supplementary Data 7, 8). Across the genomes, the H1-, H2-, and H3-T6SS apparatus gene clusters are encoded at the same respective genomic locus relative to the next adjacent non-T6SS gene of the core genome (Fig. 3a) and genetic differences between the strains' T6SS apparatus genes correlate with the phylogenetic distance between the

strains' genomes (Supplementary Fig. 5), strengthening conclusions on the vertical inheritance of the clusters since the MRCA of the species. Our stochastic mapping also showed 38 events of putative gene loss among the H1-, H2-, and H3-T6SS apparatus gene clusters (Fig. 3b). A comparison of closely related genomes showed that the missing H1- and H3-T6SS apparatus gene clusters are part of large genomic deletions of over 100 kilobases (kb), whereas the deletion of the H2-T6SS apparatus gene cluster affects almost exclusively T6SS genes (Fig. 3c and Supplementary Fig. 6).

Despite the differences in gene synteny between the four apparatus gene clusters, the individual apparatus genes are very likely to share a distant common ancestry. Two previous studies by Bingle et al.[29] and Barret et al.[30] proposed that the gene clusters were likely acquired independently from each other by horizontal gene transfer. Our data suggests that this happened for the H1-, H2-, and H3-T6SSs in the evolutionary history prior to the divergence of the species *P. aeruginosa*. To investigate this further, we complemented our study with an analysis of the clusters' GC content. The GC% content of the 12 apparatus genes (*tssA, tssB, tssC, tssE, tssF, tssG, tssJ, tssK, tssL, tssM, clpV,* and *hcp*) of the four T6SSs differs by at most 6% and is most similar between the H1- and H3-T6SS (Fig. 3d). This dissimilarity in GC% content highlights the likely origins of these clusters from interspecies transfers rather than duplication in a *P. aeruginosa* strain. When comparing the GC% content of each cluster to the genome GC% content, the H2-T6SS (average GC%: 65.72) is most similar to the genome (average GC%: 66.24). This suggests ancient acquisition and GC content refinement. In contrast to the two aforementioned studies, we observed the H1- and H3-T6SSs to have a ~3% higher GC% content than the genome GC% content suggesting more recent acquisition of these apparatus clusters and that they are yet to undergo genomic refinement. The GC% content of the H4-T6SS (average GC%: 63.24) also differs from the genome GC% content (average GC%: 66.24), supporting a hypothesis that this cluster is recent and has yet to undergo amelioration. Further, the H4-T6SS stands out by having a lower GC% content, whilst the H1- and H3-T6SSs have higher GC% content compared to the genome, suggesting different sources of the apparatus gene clusters.

Unlike for the H1-, H2-, and H3-T6SSs, we observed gain events of the H4-T6SSs since the MRCA of the species in our stochastic mapping (Fig. 3b). These events were observed at two genomic loci (Fig. 3a, e). The gene clusters are present in larger insertions of at least 60 kb (Supplementary Fig. 7) that contain non-T6SS genes of mobile genetic elements like *traG*, which encodes a protein required for conjugative transfer and could support the movement of this gene cluster by horizontal gene transfer. To probe the inheritance of the H4-T6SS apparatus gene cluster by vertical and horizontal gene transfer, we built a phylogenetic tree based on the *tssB4* sequences and the core genome of the strains (Fig. 3f). We first identified the closest related nucleotide sequences to the *P. aeruginosa tssB4* in the NCBI database and added these to the trees (Supplementary Data 9). The gene tree reveals four distinct clades of *tssB4* sequences (Fig. 3f). Indeed, *tssB4* nucleotide sequences differ much more between strains than the *tssB1, tssB2* or *tssB3* sequences (Supplementary Fig. 8a). Some of these clades are more related to *tssB* sequences from other species than to other *tssB4* sequences in *P. aeruginosa*, suggesting the movement of these genes by horizontal gene transfer between strains of different species (Fig. 3f). By overlaying the trees with the gain events predicted by stochastic mapping, we propose four independent acquisition events of *tssB4* sequences of *P. aeruginosa* strains from bacteria outside the species (Fig. 3f and Supplementary Fig. 8a, b). This evidence supports acquisition from organisms within the genus, such as *Pseudomonas chlororaphis*, as well as from outside of the genus, such as *Paraburkholderia hayleyella* based on the currently available sequence information. Further, we propose subsequent inheritance by vertical

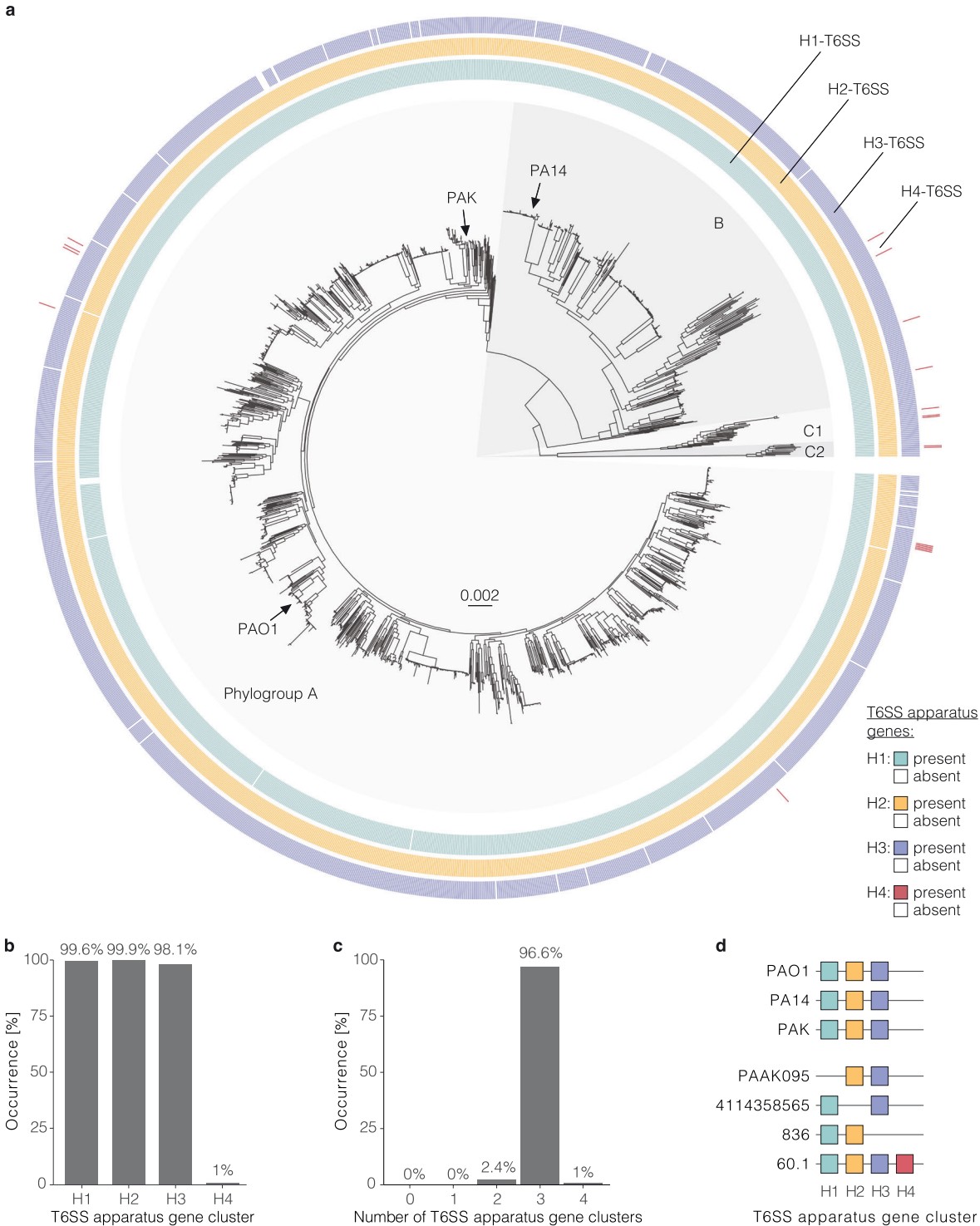

**Fig. 2 | Occurrence of the H1-, H2-, H3-, and H4-T6SS apparatus gene clusters in a global population of *P. aeruginosa*. a** Phylogenetic tree of 1960 *P. aeruginosa* strains based on a core-genome alignment. Phylogroups A, B, C1, and C2 are highlighted with grey shading and the position of three laboratory reference strains is indicated. Four outer circles indicate the presence and absence of H1-, H2-, H3-, and H4-T6SS apparatus gene clusters. The presence of the genes is indicated in colour and their absence or interruption in white. This maximum-likelihood tree was inferred using the HKY + F + I model and is midpoint rooted. Genetic distance is shown in substitutions per site. **b** Occurrence of the four T6SS apparatus gene clusters among *P. aeruginosa* strains of the dataset. **c** Occurrence of the indicated number of T6SS apparatus gene clusters in one genome among strains. **d** Schematic of the presence and absence of the four T6SS apparatus gene clusters among three commonly used laboratory reference strains (PAO1, PA14, PAK) and representative isolates of the global *P. aeruginosa* population.

gene transfer, as observed among strains PA-W1 and O14-2A after the inferred acquisition event in their common ancestor (Fig. 3f), and by lateral gene transfer between strains of the species (Fig. 3f and Supplementary Fig. 9). This lateral transfer is supported by the spread of highly similar *tssB4* sequences among distantly related *P. aeruginosa* strains of the same and different phylogroups, as observed by closely related *tssB4* sequences among distantly related strains of phylogroup A and B (Supplementary Fig. 9).

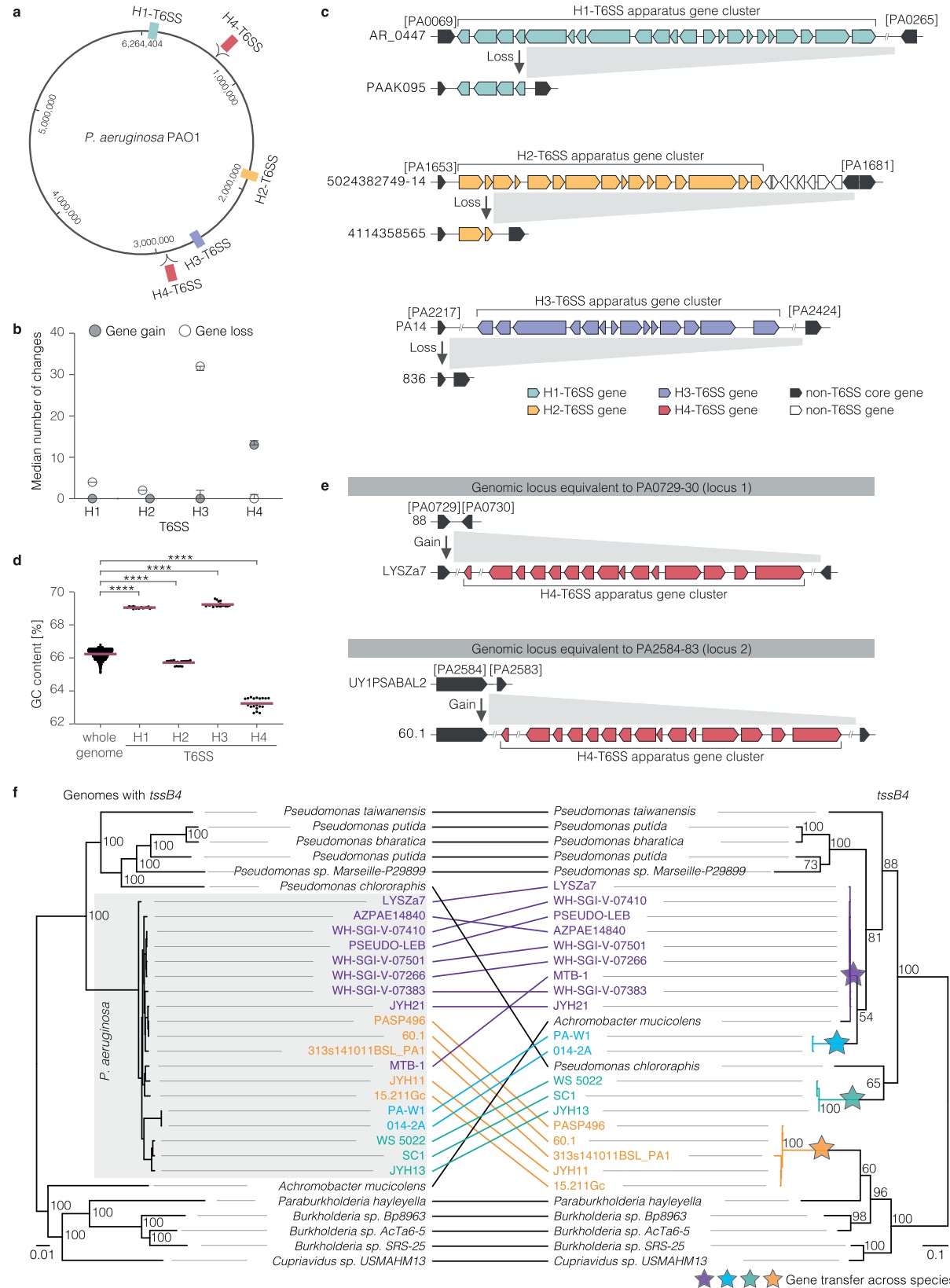

Taken together, we have found indications for the gain and loss of T6SS apparatus gene clusters in both the distant and recent evolutionary history of the species, which contributes to the clusters' observed presence and absence among today's strains.

## Genomic organisation of known effector genes of the H1-, H2-, and H3-T6SSs in the *P. aeruginosa* reference strain

Effector genes encode effector proteins that are transported physically attached to or as additional domains of various structural proteins of the respective T6SS apparatus (Fig. 4a). After their secretion, the

**Fig. 3 | Gain of H4-T6SS apparatus gene clusters from outside of the species and example of H1- to H3-T6SS loss during species diversification. a** Schematic indicating the genomic positions of the three T6SS apparatus gene clusters in *P. aeruginosa* strain PAO1. The equivalent of the genomic loci at which other strains gain the H4-T6SS apparatus gene cluster are indicated in red. **b** Gain and loss of T6SS apparatus gene clusters during the evolutionary history of *P. aeruginosa* as determined by stochastic mapping. Dots indicate the mean number of gene gains (grey) and losses (white) over 1000 simulations. Error bars indicate the lower and higher 95% confidence interval. **c** Schematic indicating the loss of H1-, H2-, or H3-T6SS apparatus genes in exemplary strains. Locus tags of the corresponding genes in the reference strain PAO1 are provided in brackets for orientation. The depicted strains with the gene cluster are close relatives of the strains without the respective gene cluster. **d** Dot plot indicating the GC% content of the whole genome of all strains in the dataset (*n* = 1960) and of the four T6SS apparatus gene clusters of the

strains in the dataset that encode four T6SSs (*n* = 19). Each dot indicates one strain. The red line indicates the mean. Statistical significance was tested by first performing a one-way ANOVA analysis (Kruskal-Wallis test) followed by a multiple comparison (Dunn's multiple comparison test). ****, *P* < 0.0001. **e** Schematic indicating the gain of H4-T6SS apparatus genes in two different genomic loci. **f** Co-phylogenetic plot of a genome-based tree of *P. aeruginosa* strains with *tssB4* and representative strains of other species with the closest related *tssB4* (left) and a *tssB4* gene tree (right). The four colours indicate *P. aeruginosa* strains that carry a similar *tssB4* gene upon independent acquisition events from bacteria outside of the species. The genome tree is a maximum-likelihood tree inferred with the HKY + F + I model. The gene tree is a maximum-likelihood tree inferred with the TN + F + I + G4 model. Ultrafast bootstrap approximations are indicated. Both trees are midpoint rooted, and their distances are shown in substitutions per site.

effectors engage in diverse activities ranging from nutrient-binding to toxicity towards other cells. So far, 22 effector proteins are known for the reference strain PAO1[3,9,12,14,16,20,24,30–51] (Supplementary Data 1). Each effector is associated with one of the three T6SS apparatuses it is secreted by. In three instances in PAO1, individual effector proteins have a dual function as a structural component and an effector (VgrG2b, Tse6 and Tse7). For instance, VgrG2b functions as a VgrG protein by forming a spike trimer on top of the Hcp tube but also has an enzymatically active C-terminal domain[42–44].

Focusing our attention on the genomic loci of effector genes, we observed their distribution across the genome of *P. aeruginosa* as depicted for PAO1 (Fig. 4b). Additional genomic loci with T6SS effector genes are known in other *P. aeruginosa* strains (Supplementary Fig. 10). In this work, we define the four different types of genomic organisations observed in which effector genes occur (Fig. 4c). Type I effector genes are surrounded by non-T6SS genes in the genome (e.g., *azu*) (Fig. 4c, d). Type II effector genes occur in a pair with an immunity protein-encoding gene (e.g., *tse1* and *tsi1*) and are surrounded by non-T6SS genes. Immunity proteins inactivate the cognate effector, which has anti-prokaryotic activity, and prevent fratricide. Type III effector genes are encoded directly adjacent to a few additional T6SS structural genes (e.g., *vgrG3* and *tseF*) and no immunity gene. Type IV effector genes occur in a pair with an immunity protein-encoding gene and are encoded adjacent to a few additional T6SS structural genes (e.g., *vgrG1c, tse5* and *tsi5*). These additional T6SS structural genes may be *hcp, vgrG* or PAAR that encode T6SS structural components required for export of the respective effector and are secreted themselves (Fig. 4a, c, d and Supplementary Fig. 11). In rare cases, type III or IV effector genes are additionally encoded next to apparatus gene clusters (e.g., *vgrG1a, tse6*, and *tsi6* next to the H1-T6SS) (Fig. 4b). Considering the various loci of effector genes within a single genome and their presence either as isolated genes or in the neighbourhood of other T6SS genes made us wonder about the occurrence of effector genes in phylogenetically diverse strains of the species.

**Pan-genomic analysis of H1-, H2-, and H3-T6SS effectors reveals core and accessory effectors in the global *P. aeruginosa* population**

To test for the intraspecific distribution of effector genes, we analysed their occurrence in our dataset of diverse *P. aeruginosa* genomes from around the world. To avoid a bias because of the lack of a T6SS apparatus gene cluster in some strains, we analysed only those genomes of the dataset with H1-, H2- and H3-T6SS apparatus genes (*n* = 1912). Effectors of the H4-T6SS were omitted here as none have been characterised to date. We found some effector genes widely distributed and some others with patchy distribution (Fig. 5a and Supplementary Data 10–12). When analysing all effector genes by occurrence, we noticed a stark difference between fifteen widely distributed genes present in 95–100% of strains and the remaining genes present in only a fraction of the strains, confirming previous results on

much smaller datasets of local bacterial populations[25,26]. This finding made us think of the pan-genome concept that distinguishes between core (present at a high frequency in a species) and accessory (found in a fraction of strains) genes. This concept has not yet been applied to the T6SS and is dependent on a large dataset to be tested for validity. We applied the established threshold of 95% occurrence for genes of the soft-core genome[52] to T6SS effector genes in our dataset (Fig. 5b). Accordingly, we classify the *P. aeruginosa* effectors into six core and nine accessory effectors of the H1-T6SS, seven core and ten accessory effectors of the H2-T6SS, and two core and three accessory effectors of the H3-T6SS (Fig. 5c). This finding was recapitulated when analysing the occurrence of the effector genes only in the complete genomes of the dataset with all three apparatus gene clusters (*n* = 232) (Supplementary Fig. 12). Taken together, we have identified core and accessory effectors in a global population of *P. aeruginosa* strains.

**T6SS effector genes of the *P. aeruginosa* most recent common ancestor**

To explore the intraspecies evolutionary history of T6SS effectors in *P. aeruginosa* strains, we reconstructed the presence of effector genes in the MRCA of the species. Therefore, we built a phylogeny of the *P. aeruginosa* strains with its closest related species *Pseudomonas paraeruginosa* (formerly considered part of the *P. aeruginosa* species[53]) as a separate branch and seven representative sequences of other closely related species as an outgroup (Fig. 6a, Supplementary Fig. 13 and Supplementary Data 13–15). Stochastic mapping was used to infer the effector genes in the MRCA based on the presence and absence of each effector gene at the tips of the tree. This method determines the likelihood of the presence or absence of an effector gene at each node along the branches of the tree. The MRCA harboured eight effector genes of the H1-T6SS, eight effector genes of the H2-T6SS, and three effector genes of the H3-T6SS (Fig. 6b, Supplementary Fig. 14 and Supplementary Data 16, 17). The core effector genes were all present in the MRCA. However, not all effector genes of the MRCA are core effectors in our dataset of diverse strains, which made us wonder about the loss of effector genes in the species' evolutionary history. Multiple effector genes of the reference strain PAO1 and other strains were not in the species' MRCA, which made us curious about the strains' gain of effector genes during the diversification of the species.

**Intraspecies inheritance of effector genes by vertical and lateral transfer among *P. aeruginosa* strains**

The difference in effector genes between the *P. aeruginosa* MRCA and present strains suggested the gain and loss of effector genes during the diversification of the species. Multiple effector genes like *tas1, tse7, tseV*, and *tepB* were present in the MRCA but absent from a substantial number of today's strains, suggesting gene loss in sub-populations. Vice versa, some effector genes like *tse6, tle4b*, and *rhsP2* were not present in the MRCA and expected to be acquired in the more recent evolutionary history of the species. To quantify these events in

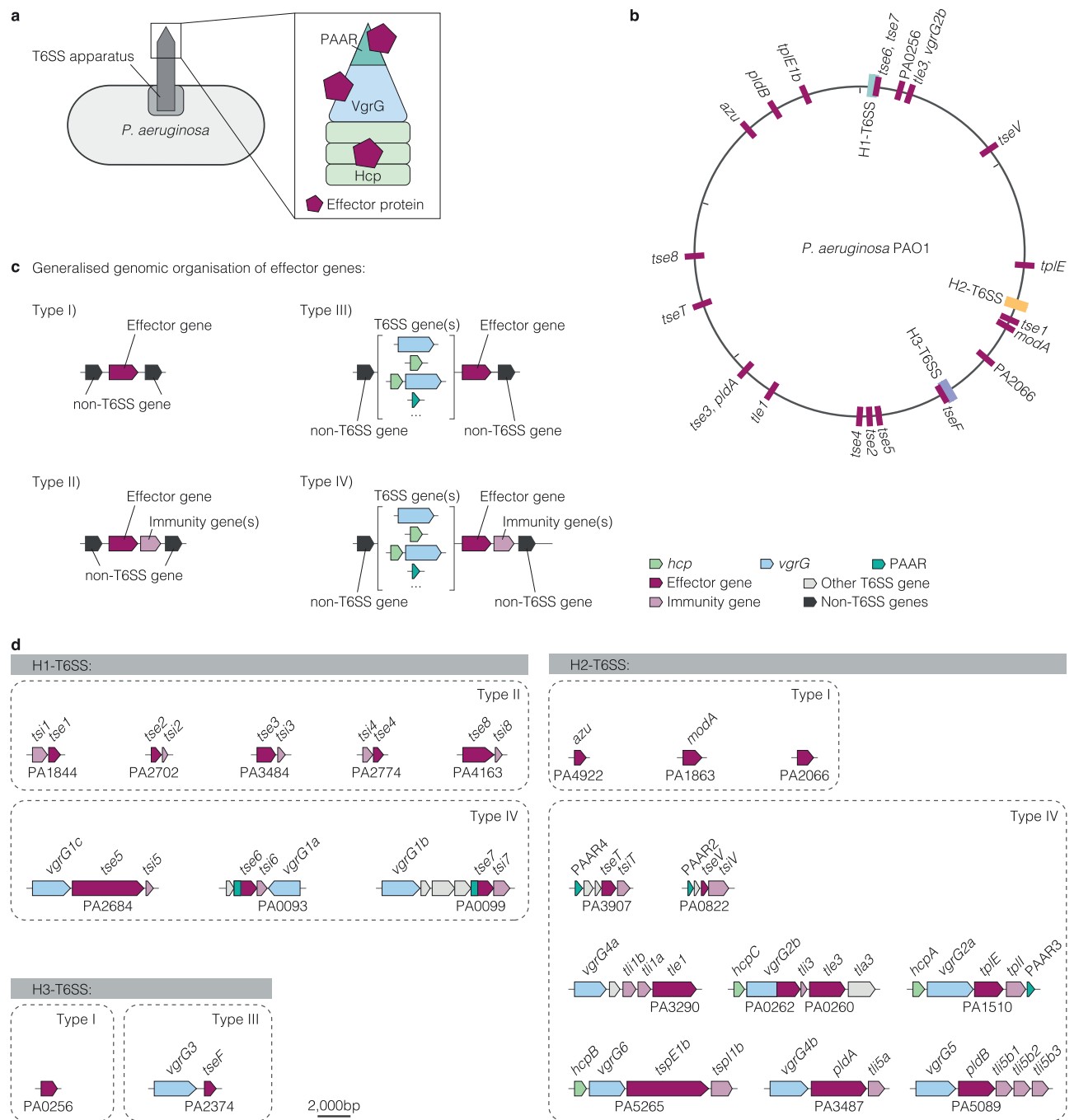

**Fig. 4 | Organisation of H1-, H2- and H3-T6SS effector genes in the PAO1 genome and homology of *vgrG* and *hcp* genes. a** Schematic of T6SS effector protein secretion. Each effector is associated with either a Hcp, VgrG or PAAR protein of the T6SS apparatus for secretion. **b** *P. aeruginosa* reference strain PAO1 encodes 22 effector genes. Schematic of the PAO1 genome indicating the position of the individual genes (in maroon). Genes are labelled with their name or locus tag if no name is available. **c** Generalised schematic of four different types of genomic organisation of effector-encoding genes we define based on the adjacent genes of an effector gene: Type I) effector gene with no immunity gene(s) or T6SS structural gene(s), type II) effector gene with immunity gene(s), but with no T6SS structural gene(s), type III) effector gene without immunity gene(s), but with other T6SS structural gene(s), type IV) effector gene with immunity gene(s) and other T6SS structural gene(s). **d** Graphical depiction of the genomic organisation of the 22 effector gene loci in the reference strain PAO1. Dotted lines group effector genes by the type of their genomic organisation.

descendants of the MRCA, we used stochastic mapping to reconstruct the evolutionary history of each effector gene on each node of the species phylogenetic tree. We identified up to 100 events of gain or loss of individual effector genes during the diversification of the species (Fig. 7a). These events are primarily observed among accessory and not core effector genes. When looking at the relative number of gain and loss events of individual genes, some are found more often to be acquired than deleted. A strong example is *tle2,* which is more often acquired than lost. Vice versa, *tseV* is more often lost than gained. The gene *rhsP2* is both gained and lost, with the frequency of gene gain being slightly higher.

To further investigate the source of the observed intraspecific diversity in effector genes, we performed a detailed phylogenetic analysis of the effector genes. For space reasons, we focus on *rhsP2* as

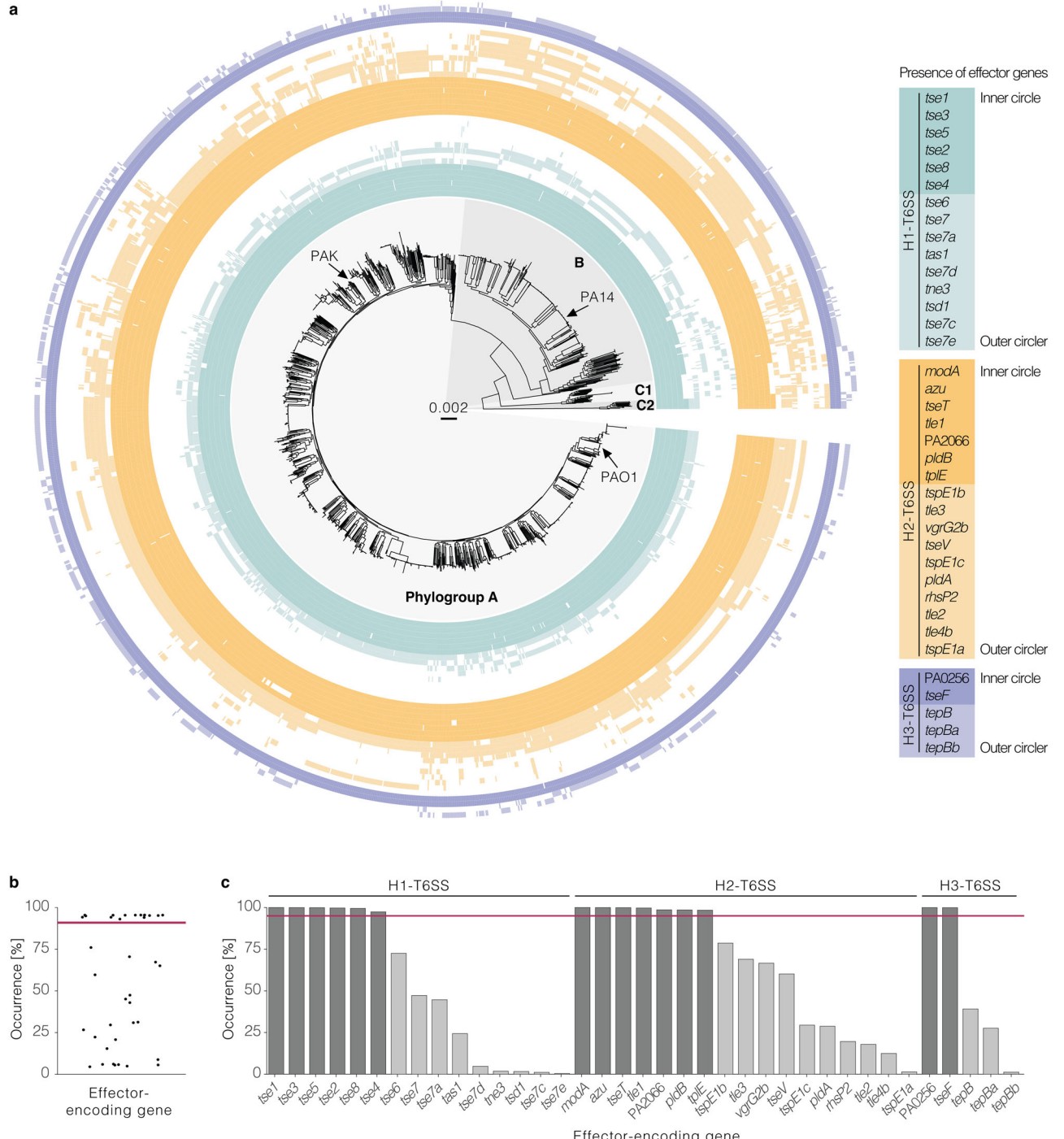

**Fig. 5 | Distribution of T6SS effector genes across strains of a global *P. aeruginosa* population reveals core and accessory effectors. a** Maximum-likelihood phylogenetic tree of 1912 *P. aeruginosa* strains based on a core genome alignment. Coloured circles indicate the presence of the indicated effector genes. The tree was inferred using the HKY + F + I model and is midpoint rooted, the genetic distance is shown in substitutions per site. A list of the strains and their effector genes is provided in Supplementary Data 10. **b** 15 out of 37 effector genes occur in more than 95% of the analysed strains (red line). Each dot indicates an effector gene. **c** Occurrence of the indicated effector genes in *P. aeruginosa* genomes of the dataset. The horizontal line indicates an occurrence of 95%. Dark shades indicate the presence in the core genome, whilst light shades depict those in the accessory.

an example in the main text and provide data for the remaining effector genes in supplementary files (Supplementary Data 18–21). The effector gene *rhsP2* is particularly interesting as it is found with a patchy distribution among strains of three phylogroups A, B, and C1 (Fig. 5a) but absent from the MRCA of *P. aeruginosa* and the MRCAs of these phylogroups (Fig. 6b and Supplementary Fig. 14). This effector originally arose by fusion of a domain of YD repeats to a C-terminal

domain with toxic ADP-ribosyltransferase activity[54,55] and homologues of *rhsP2* are also found in strains of other species like *Pseudomonas chlororaphis* and *Pseudomonas viridiflava* (Supplementary Data 18, 19). To test for lateral and vertical transfer of *rhsP2* that could explain the above-mentioned gain and loss events, we built a phylogenetic tree based on the effector gene and a tree based on the core genome of the strains harbouring this effector gene (Fig. 7b and Supplementary

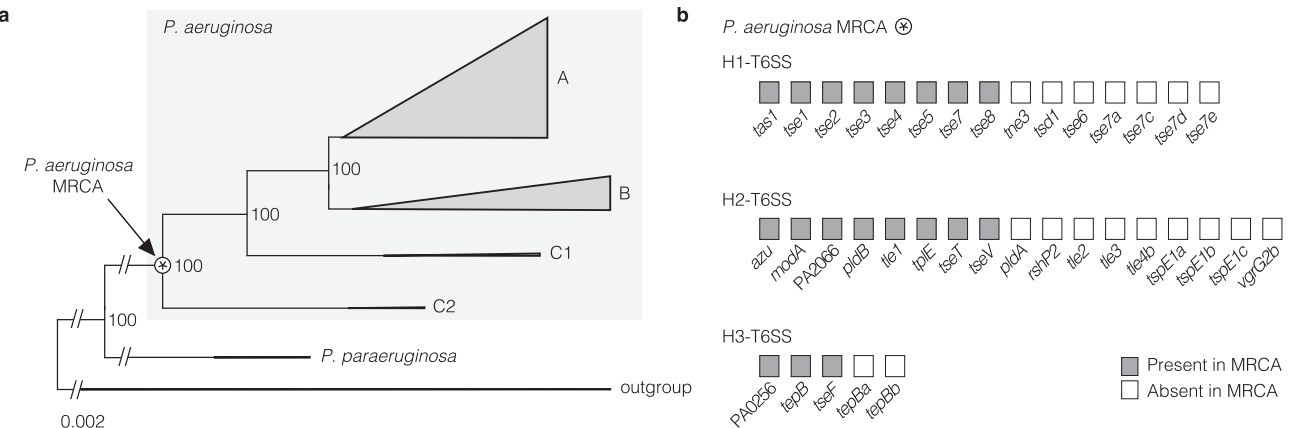

**Fig. 6 | Ancestral reconstruction predicts presence or absence of T6SS effector genes in the *P. aeruginosa* most recent common ancestor (MRCA). a** Phylogeny based on a core-genome alignment *of P. aeruginosa* (n = 1912) and *P. paraeruginosa* (n = 5). Seven strains of the species *P. delhiensis, P. knackmussii, P. humi, P. jinjuensis, P. multiresinivorans, P. nitroreducens,* and *P. panipatensis* were used as an outgroup.

The maximum-likelihood tree was inferred with the HKY + F + I model and is rooted to the outgroup. Branches were collapsed. The names of the phylogroups are indicated. Ultrafast bootstrap values are indicated. **b** 19 effector genes are detected in the MRCA of *P. aeruginosa* based on stochastic mapping.

Fig. 15). We observed that *rhsP2* of *P. aeruginosa* strains are more closely related to each other than to *rhsP2* of other species, suggesting a single acquisition event of *rhsP2* by an ancestral *P. aeruginosa* strain and subsequent inheritance among strains of the species. We observed intraspecies inheritance by vertical and lateral transfer and the spread of *rhsP2* to strains in other clades of phylogroup B and to a few strains of phylogroup A (Fig. 7c and Supplementary Fig. 16). As an example of intraspecies lateral transfer, *rhsP2* with identical nucleotide sequences is found in phylogenetically diverse strains of phylogroup A following three independent acquisition events (Fig. 7c, d). As an example of vertical transfer following an acquisition event, the topologies of the gene tree and species tree cladding (see the blue lines) match for strain MRSN 317 and its clade (Fig. 7c, e). By specifically focusing on MRSN 317 and the *rhsP2* cladding, we see additional evidence of sequence diversity present in both the gene and genome of this strain compared to the rest of the blue clade as well as an inferred loss event (Fig. 7e). Similar observations are made for the other effector genes (Supplementary Data 20, 21). Taken together, we determined single gain events of effector genes by *P. aeruginosa* strains, most likely via horizontal gene transfer from strains of other species, and subsequent intraspecies inheritance by vertical and horizontal transfer.

### Transfer of effector genes in T6SS genomic islands by homologous recombination and a potential role of transposons

To investigate the mechanisms of transfer, we focused on the genomic loci that harbour accessory effector genes. For each gene, we assessed whether it is located at the same locus across acquisition events among the analysed genomes and if there are remnants of mobile genetic elements present. Continuing with *rhsP2* as an example, we found *rhsP2* at the same locus (equivalent to PA14_43100 in the strain PA14) in between the H2-T6SS apparatus gene cluster and the same non-T6SS gene of the core genome that is present across strains and across inferred acquisition events (Fig. 8a i). When comparing the genomic locus upon gain and loss of *rhsP2* as shown for three closely related strains (PA_151970, BWHPSA043, PPF-13), we observed the inferred gain and loss of *rhsP2* together with its four adjacent T6SS genes that encode proteins for the effector's secretion (*hcp2, vgrG14,* and *tap14*) and immunity (*rhsI2*) (Fig. 8a ii–iv and Supplementary Fig. 17). This data suggests transfer of an entire small T6SS genomic island of multiple T6SS genes between strains. One strain (AR_0457) additionally encodes a gene for a transposase of the IS3 family next to *rhsP2* and provides a hint for a potential mechanism of effector acquisition by

transposition (Fig. 8a v). Genes encoding transposases of the same family were previously observed next to effector genes in multiple bacterial species[25,56]. Five other accessory effector genes (*tle4b, tseV, pldA, tle2, tspE1a*) undergoing gain/loss were also found exclusively at one locus each (Fig. 8b). However, no transposase was observed next to these effector genes in any of the analysed strains. This would argue for the acquisition of the genetic element from other strains by homologous recombination. However, we did not detect conserved recombination breakpoints upstream and downstream of the acquired element in sequence alignments, which is not uncommon and can be explained by variation in the exact breakpoints between transfers and time since these events occurred[57,58]. In addition, all five of these genes encode H2-T6SS effectors and have a type IV genomic organisation. Four are within a T6SS genomic island with a *vgrG* gene (*tle2, tle4b, pldA, tspE1a*) and one in a T6SS genomic island with a PAAR gene (*tseV*) that would have likely been acquired as a gene block supporting homologous recombination of the conserved non-T6SS flanking genes between *P. aeruginosa* strains.

Among all the effector genes, we observed only one case of the two adjacent effector genes *vgrG2b* and *tle3* encoded at one locus by some strains and at another locus by others. Most strains encode these genes at a locus equivalent to PA0260 and PA0262 in the reference strain PAO1 together with three other T6SS genes and a transposase-encoding gene of the IS3 family (Fig. 8c and Supplementary Fig. 18a). As observed for *rhsP2*, this element including *vgrG2b* and *tle3* is repetitively gained and lost at this locus, suggesting transfer by homologous recombination and potentially by the transposase. In specific strains of the phylogroup C1, we observed *vgrG2b* and *tle3* at a different locus (neighbouring non-T6SS core genes equivalent to PA3858 and PA3859 in PAO1) (Fig. 8d i). Stochastic mapping inferred one single gain event of *vgrG2b* and *tle3* at the second locus and four loss events resulting in a patchy distribution of *tle3* and *vgrG2b* in phylogroup C1 strains (Fig. 8d ii–iv, Supplementary Fig. 18b and Supplementary Data 22). Although transposon-mediated transfer could explain an acquisition at a different site, we did not find a transposase next to the effector genes at the second locus. Further, the nucleotide sequences of *tle3* of the second locus form a separate branch in the gene tree than the *tle3* sequences of the first locus, which makes two independent acquisition events from outside the species more likely than a transfer between loci from one strain to another (Supplementary Data 20). Other strains of the phylogroup C1 encode the effector genes at the first locus following an inferred gain event (Fig. 8d iii). Phylogenetic

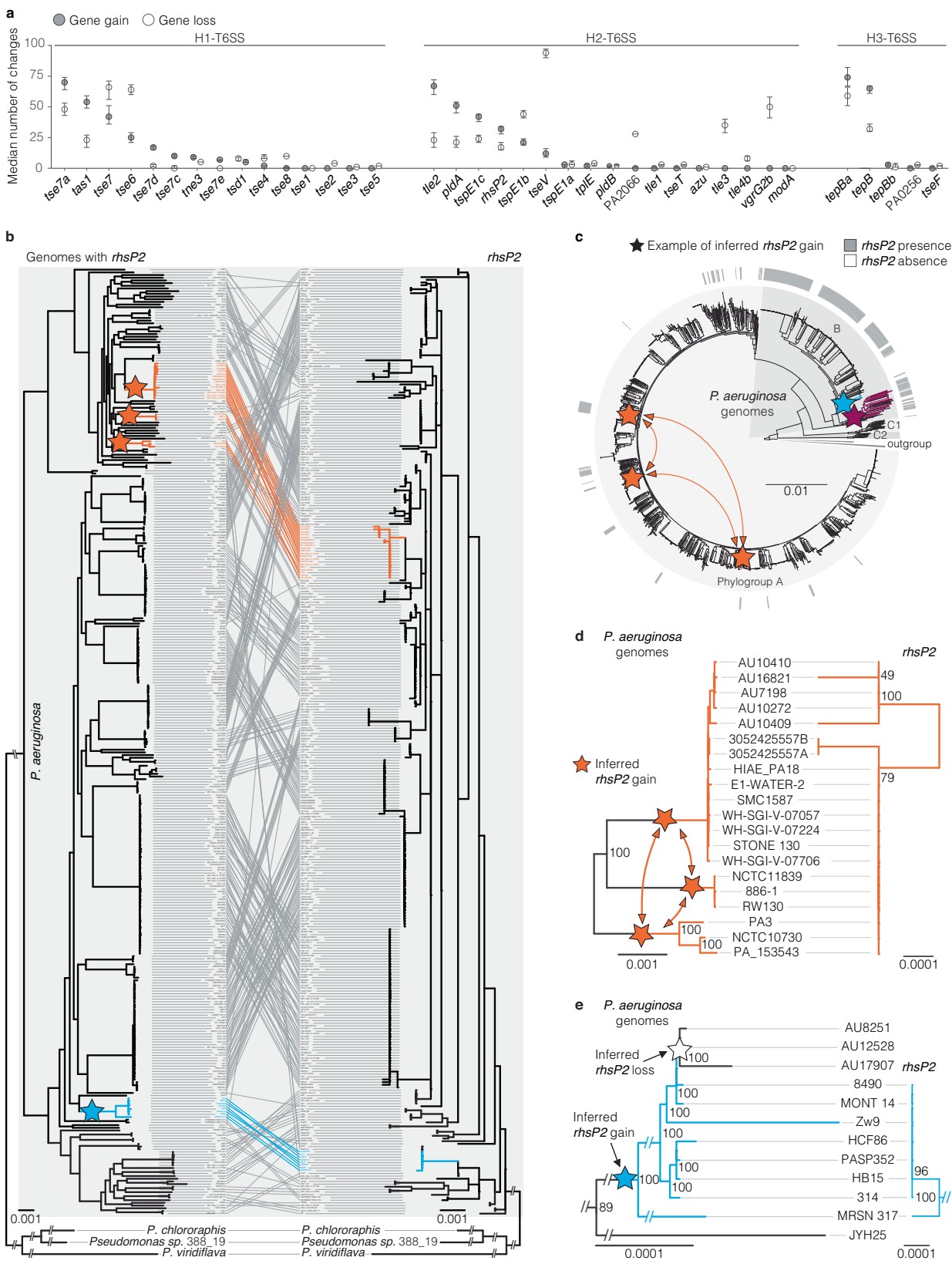

**a** Gene gain ● Gene loss ○

H1-T6SS · H2-T6SS · H3-T6SS

**b** Genomes with *rhsP2* | *rhsP2*

*P. aeruginosa*

*P. chlororaphis* / *P. chlororaphis*
*Pseudomonas sp.* 388_19 / *Pseudomonas sp.* 388_19
*P. viridiflava* / *P. viridiflava*

**c** ★ Example of inferred *rhsP2* gain | *rhsP2* presence / *rhsP2* absence

*P. aeruginosa* genomes

B · C1 · C2 outgroup · Phylogroup A

**d** *P. aeruginosa* genomes | *rhsP2*

★ Inferred *rhsP2* gain

**e** *P. aeruginosa* genomes | *rhsP2*

Inferred *rhsP2* loss · Inferred *rhsP2* gain

analysis of their *tle3* revealed high similarity to *tle3* of phylogroup A strains (Supplementary Data 20), suggesting horizontal gene transfer across strains of different phylogroups. Of note, we did not find genomes that harboured *tle3* and *vgrG2b* at both loci simultaneously. Taken together, we identified putative mechanisms by which the effector genes are gained and lost as part of genomic islands.

## Frequent swapping of effector genes at four genomic loci by recombination

At four genomic loci (equivalent to PA0093, PA0099, and PA5265 in reference strain PAO1 and PA14_33970 in reference strain PA14), we observed distinct effector genes at the same site across strains (Fig. 8e, f). Taking the locus that harbours either *tspE1b* or *tspE1c* as an

**Fig. 7 | Dynamic gain and loss of T6SS effector genes during diversification of the species. a** Gene gain and gene loss events during the evolutionary history of *P. aeruginosa* based on stochastic mapping. Dots indicate the mean number of gene gains (grey) and losses (white) over 1000 simulations. Error bars indicate the lower and higher 95% confidence interval. **b**–**e** Gain of an effector gene, *rhsP2*, by *P. aeruginosa* and subsequent vertical and lateral transfer between *P. aeruginosa* strains (examples for transfer events are highlighted in the same colour across panels). **b** Co-phylogenetic plot of the species tree on the left and the *rhsP2* tree on the right. Lines in the middle connect sequences from the same genome. The figure shows that *rhsP2* sequences of *P. aeruginosa* strains are more closely related to each other than to *rhsP2* genes of other species, suggesting a single acquisition event of the gene and subsequent intraspecies inheritance. The genome tree was built on the core genome of *rhsP2* + *P. aeruginosa* genomes and three strains of related species and calculated using the HKY + F + I model. The gene tree is based on a multiple sequence alignment of *rhsP2* from these strains using the TPM2 + F + R2

model. Both trees are maximum-likelihood trees and midpoint rooted. A full-size image of the co-phylogeny without cut branches is shown in Supplementary Fig. 15. **c** Intraspecies inheritance of *rhsP2* by vertical and horizontal transmission. Maximum-likelihood phylogenetic tree of the *P. aeruginosa* strains in our dataset. The outer ring shows the presence (grey) or absence (not filled) of *rhsP2* in the respective genomes. Stars denote effector gene acquisition events predicted by stochastic mapping, and arrows indicate the inferred transfer of the effector gene between the ancestors of the sequenced strains. The purple star is referred to in Fig. 8a. The tree was computed using the HKY + F + I model and midpoint rooted. **d** Focused view of the orange example showing three diverse clades (note other branches removed, see b/c) all encode a highly similar *rhsP2* gene evidencing likely horizontal gene transfer events of *rhsP2* between ancestral *P. aeruginosa* strains (orange arrows). **e** Focused view of the blue example in which the topology and diversity of the effector gene (*rhsP2*) follows that of the species supporting vertical transmission. Ultrafast bootstrap values are indicated.

example (Fig. 8e), no effector is predicted at this site in the MRCA (Fig. 6b). By analysing this locus among closely related strains, we infer the acquisition of a T6SS genomic island that contains *tspE1c* and four other T6SS genes as it is found in strain AUS128 (Fig. 8e iii, iv and Supplementary Fig. 19–21). The other T6SS genes in the element, such as *hcp*, *vgrG,* and, if present, the *tap* gene, encode proteins that aid in the secretion of the effector. Strain PA233 shares the same common ancestor as AUS128 and encodes a similar T6SS island with the same synteny of genes, except that the 3'end of *vgrG6*, effector gene *tspE1c*, and immunity gene *tspI1c* were replaced with a newly gained and different; 3' end of *vgrG6* (that when expressed likely mediates specific binding and loading of TspE1b but not TspE1c)*, tspE1b*, and *tspI1b* (Fig. 8e iv). TspE1b and TspE1c are too divergent to be properly aligned (Supplementary Fig. 19a) and are expected to differ in their pyocin-like effector activity[25]. We also see clear evidence of the loss of the effector gene and most of the genetic element or exchange back to the *tspE1c* cluster in inferred descendant strains with shared common ancestry to strains JB2, PA233 and AU10014 (Fig. 8e iii). The transition state diagram based on stochastic mapping suggests that such loss of effector genes is rather rare at this locus, and the exchange between *tspE1b* and *tspE1c* is much more common among strains of the species (Fig. 8e ii and Supplementary Data 23). To test for recombination within this genetic element, we applied the recombination detection tool RDP5 and identified multiple recombination breakpoints (Fig. 8e v). We observed strong data supporting one dominant break-point at the 3'end of the *vgrG* gene that is encoded in direct vicinity to the effector gene and less precise predicted recombination sites downstream (e.g., in *argH*/PA5263) of the effector gene. These findings are consistent with homology-facilitated illegitimate recombination, which leads to the acquisition of a DNA element by recombination at homologous and non-homologous sites[59]. Similarly, the gain of an effector gene by the replacement of an existing one is observed at three additional genomic loci (equivalent to PA0093, PA0099 in reference strain PAO1 and to PA14_33970 in reference strain PA14) (Fig. 8f). At these loci, we also find indications for gene swapping by homologous recombination (Supplementary Fig. 22–24). The proteins encoded by different effector genes at these loci are also divergent in their enzymatically active domain, as previously shown for two effector proteins encoded by genes at the PA0093 locus: Tse6 with experimentally characterised NAD(P) + glycohydrolase activity[60] and Tas1 with experimentally characterised (p)ppApp synthetase activity[23]. At the PA0099 locus, a different alphabetic letter was used for each effector gene at this locus that encodes a protein with a divergent catalytic domain and less than 30% amino acid identity in this domain with other effector proteins encoded at this site (Supplementary Fig. 23a and Supplementary Data 24, *tse7b* has meanwhile been relabelled to *tsd1*). In sum, we found clear evidence of frequent swapping of effector genes at four loci during species diversification and demonstrated homology-facilitated

illegitimate recombination and homologous recombination as mechanisms that enable switching.

## Gain, loss, and exchange of individual effector genes generate high intraspecific diversity of effector sets

Next, we investigated the co-occurrence of effector genes that arose in the genomes by inheritance from the species' MRCA and subsequent gain, loss, and exchange of individual effector genes. The combination of effector genes associated with one T6SS apparatus in a given strain is referred to as an 'effector set'. We had three reasons for this analysis: First, multiple different effector proteins are secreted simultaneously with one shot of the secretion apparatus they are associated with, and there might be steric hindrance or synergy between effectors that could affect the co-occurrence of effector genes in a genome positively or negatively over time. Second, the T6SS apparatuses differ in their regulation, and the strategic use of the individual apparatus might be reflected in the diversity of the effector genes that are associated with the apparatus. Third, most effectors play a role in bacteria-bacteria killing and, in particular, intraspecies killing, thus the effectors that two strains attack each other with, and their presence in the genome, serves as a proxy for the ability of two strains to co-exist or kill one another.

In the H1-T6SS, we observed co-occurrence of *tse1, tse3,* and *tse4* in the species' MRCA and core genome (Fig. 9i, ii), supporting previous findings on synergies between the effectors encoded in these genes[33]. No statistically significant co-occurrence (Hypergeometric test, false discovery rate (FDR) = 0.05, *P* < 0.05) was observed between any pairs of accessory effectors encoded in various combinations at the loci equivalent to PA0093 and PA0099 (Supplementary Fig. 25 and Supplementary Data 25). The most common effector set (ES1) was found in 689 of the analysed strains and consisted of six core effectors, the accessory effectors *tse6* at position PA0093, and *tse7* at position PA0099 (Fig. 9iii, iv). In total, we found 35 different effector sets among effector genes of the H1-T6SS, which is much fewer than the number of 1792 theoretically possible combinations of existing effector genes assuming random mix and match (Supplementary Figs. 26, 27). The effector sets of the H1-T6SS stood out by little variation in the number of effectors per set and a high number of mutually exclusive accessory effectors, which is in line with the defensive use of the H1 system in bacterial warfare as further analysed in the next section. Despite the genetic diversity between the H1-T6SS effector sets, the subcellular structures of bacteria that are targeted by the effectors after translocation (herein referred to as the prokaryotic T6SS targetome) remain the same across most sets (Fig. 9v). A similar phenomenon was observed among *Serratia marcescens* strains[61].

In the H2-T6SS, eight effector genes co-occur in the species' MRCA and core genome (Fig. 9vi, vii). Testing for the co-occurrence between pairs of accessory effectors that are encoded at different

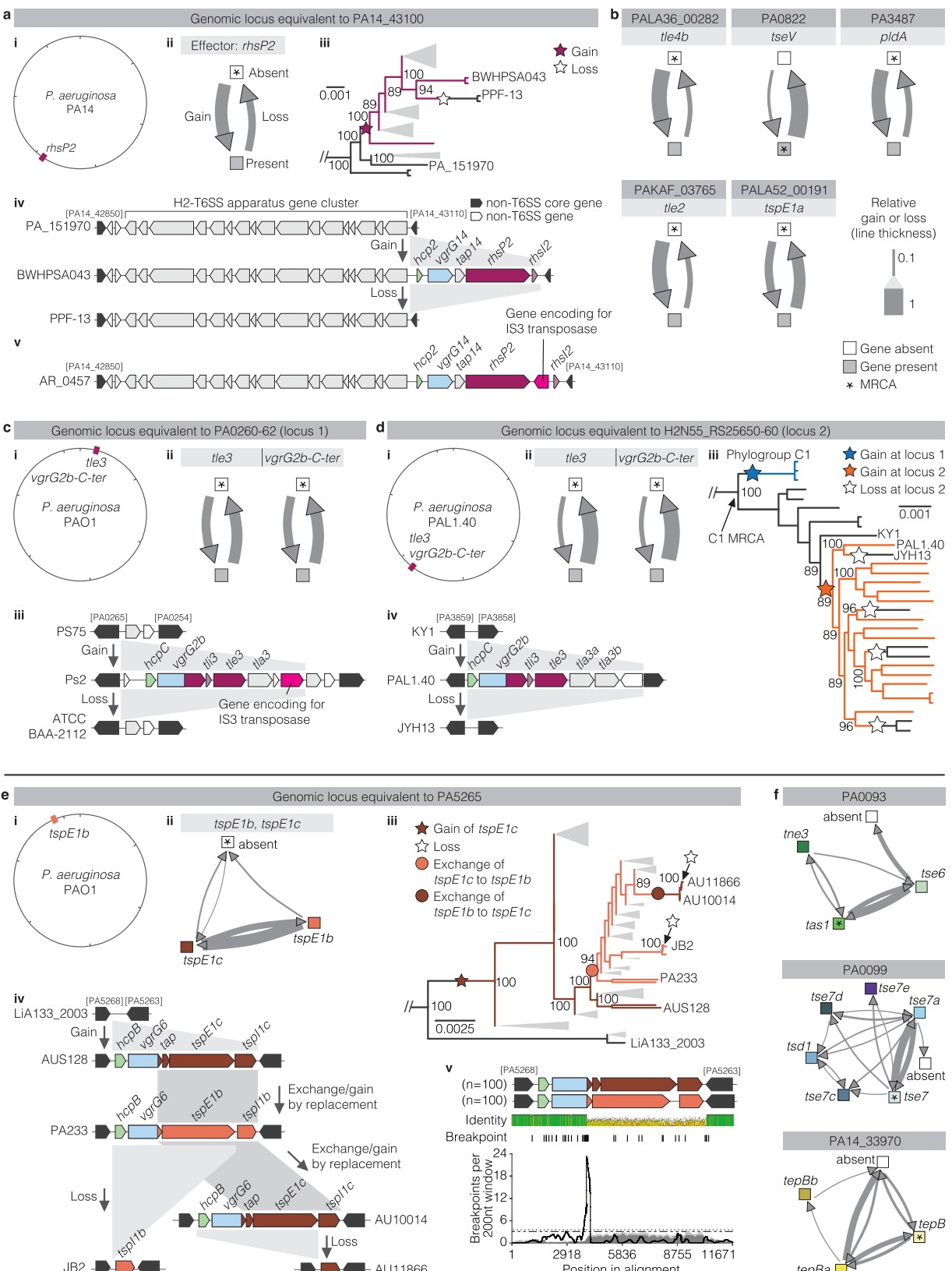

genomic loci revealed one statistically significant association (Hypergeometric test, FDR = 0.05, P < 0.05) between the effector genes *pldA* and *tseV* (Supplementary Fig. 25 and Supplementary Data 25). The effector set with the highest occurrence among the analysed genomes (n = 279) consists of the seven core effectors and three accessory effectors *tle3*, *vgrG2b*, and *tspE1b* at loci PA5265 (Fig. 9viii, ix). Out of

98,304 theoretically possible H2-T6SS effector sets, we observed only 120 different sets among the analysed genomes (Supplementary Figs. 26, 28–30). In contrast to the H1-T6SS effector sets, which differ by mutually exclusive effectors at two loci, H2-T6SS effector sets differ in only one locus (PA5265) with mutually exclusive effectors (*tspE1b* and *tspE1c*). Diversity among H2-T6SS effector sets mostly arises by the

**Fig. 8 | Genomic loci with gain, loss, and exchange of effector genes. a**, **b** Gain and loss of an effector gene at one genomic locus across strains. **a** Example of *rhsP2* at loci equivalent to PA14_43100 (strain PA14). (i) Graphical depiction of the genome of strain PA14 indicating the position of the effector gene *rhsP2*. Tick marks indicate 1Mbp. (ii) Diagram indicating the relative frequency of gain and loss events of *rhsP2* in our dataset of *P. aeruginosa* genomes. Line thickness indicates relative frequency, see key in panel (**c**). (iii) Section of the *P. aeruginosa* phylogenetic tree showing the presence (maroon lines) and absence (black) of *rhsP2*. Stars indicate gain and loss events of *rhsP2* as predicted by stochastic mapping. On all panels, distances are shown in substitutions per site, and ultrafast bootstrap values are indicated. (iv) Examples of inferred gain and loss events of the T6SS genomic island, including *rhsP2* in the ancestors of indicated strains. The depiction shows the genomic region between non-T6SS core genes equivalent to PA14_42850 and PA14_43110 in strain PA14. (v) Graphical depiction of the genomic locus encoding *rhsP2* in strain AR_0457 with the gene encoding for an IS3 family transposase highlighted in pink. **b** Diagrams indicating the relative frequency of effector gain and loss events of five additional accessory effector genes that were found at single loci only. **c**, **d** Gain and loss of a pair of effector genes at two different loci across strains. **c** (i) Graphic of the PAO1 genome indicating the position of adjacently encoded effectors *tle3* and *vgrG2b-C-ter* at the genomic locus equivalent to PA0260-62, i.e., locus 1. (ii) Diagrams showing the relative gene gain and loss frequencies of *tle3* and *vgrG2b-C-ter* at locus 1. (iii) Graphical depiction of the genomic region of locus 1 indicating an exemplary inferred gain event and an inferred loss event between the ancestral strains of those indicated. **d** (i) Graphic depicting the genome of strain PAL1.40, a strain that encodes effectors *tle3* and *vgrG2b-C-ter* at the genomic locus equivalent to locus tags H2N55_RS25650-60 in PAL1.40, i.e.,

locus 2. (ii) Diagrams indicating the relative frequencies of *tle3* and *vgrG2-C-ter* gain and loss events at locus 2. (iii) Zoom into the phylogenetic tree of the global population only showing strains of phylogroup C1. Branches are coloured based on absence (black) or presence of *tle3* and *vgrG2b-C-ter* at locus 1 (blue) or locus 2 (orange). Inferred gene gain and loss events are indicated by stars. iv) Graphical depiction of gene gain and loss events in locus 2. The genomic region between non-T6SS core genes (equivalent to locus tags PA3859 and PA3858 in PAO1) is shown. **e**, **f** Four additional examples of effector gain by replacement. **e** Example of the locus equivalent to PA5265 of the strain PAO1 across strains. (i) Graphical depiction of the PAO1 genome indicating the locus encoding effector *tspE1b*. (ii) Transition state diagram indicating the relative frequency of gene gain, loss or exchange at this locus. (iii) Part of the phylogenetic tree of the *P. aeruginosa* population highlighting examples of effector gene gain, exchange and loss at locus PA5265. iv) Examples of effector gene gain, exchange and loss in the ancestors of the indicated strains. (v) The *tspE1b* and *tspE1c* locus have a clear recombination breakpoint. The graph shows the recombination breakpoint distribution of the genomic region encoding the mutually exclusive effectors *tspE1b* and *tspE1c*. The black line indicates the breakpoint number, i.e., the number of detected recombination breakpoints within 200 nucleotides. Dark grey and light grey shaded areas indicate local 95% and 99% confidence intervals, respectively. Dashed lines indicate global 95% and 99% confidence intervals. The cartoon above indicates the genomic region. **f** Transition state diagrams indicating the frequency of gene exchange events between mutually exclusive accessory effectors at loci equivalent to PA0093 and PA0099 in PAO1, and PA14_33970 in PA14. Line thickness reflects the relative frequency as indicated in the key in (**b**).

presence and absence of individual effector genes at various genomic loci (*tle3*, *tle4b*, *vgrG2b*, *tseV*, *pldA*, *rhsP2*, *tle2*, *tpsE1a*). As a consequence, effector sets of the H2-T6SS show high variation in the number of effector genes within the set, which is not observed for H1-T6SS effector sets, resulting in a higher number of distinct sets. The prokaryotic targetome of the H2-T6SS consists of the membrane and nucleic acids across effector sets and is distinct from the targetome of the H1-T6SS (Fig. 9v).

For the H3-T6SS, effector sets consist of two co-occurring effector genes of the species' MRCA and core genome and, in some cases, an additional third effector gene at a locus of mutually exclusive effectors (Fig. 9x, xi). We observed seven different effector sets among the analysed genomes (Fig. 9xi, xiii and Supplementary Fig. 31). Although the effector sets of the H3-T6SS consist of much fewer effector genes than those of the H1-T6SS, they are similar to each other as their diversity is dominated by mutually exclusive accessory effectors at few loci. Intriguingly, the ratio of core effectors to accessory effectors is consistently 2:3 across the effector sets associated with all three T6SS apparatuses.

Next, we considered the effector genes for all three T6SSs in a strain's genome altogether, which we refer to as a combined effector set (CES). We calculated as many as 2,818,572,288 theoretically possible unique CESs in *P. aeruginosa*. In our dataset of 1912 strains, we found a total of 355 unique CESs (Supplementary Fig. 32a). The total number of T6SS effectors per genome varies from 17 to 25 (Supplementary Fig. 32b). Some CESs were present in more strains than others. The most abundant CES is found in 104 strains and contains 21 effectors including the accessory effectors *tse6*, *tse7a*, *tle3*, *vgrG2b*, *tspE1b*, and *tepB* (Supplementary Fig. 32c). Of note, this CES is distinct from the CES of the frequently studied reference strains PAO1, PA14, and PAK (Supplementary Fig. 32c). Only 20, 40, and 2 other strains have the same combination of effectors encoded by the three reference strains, respectively. Moreover, clinically relevant sequence types, including the 21 recently described major epidemic clones[62], differ from each other in their effector sets (Supplementary Figs. 33–35). To test whether additional CESs are still expected to be discovered, we performed a rarefaction curve analysis. Indeed, the curve is not saturated (Supplementary Fig. 36) and more unique CESs are expected to be found in future studies by sequencing more strains and by identifying additional effectors.

## Intraspecific effector diversity is associated with multiple effector characteristics

After observing the tremendous intraspecific diversity generated by accessory effectors, we wondered about evolutionary drivers and constraints that could explain the variation in the occurrence between effector genes. We hypothesised that differences between effector genes in their genomic organisation and between effector proteins in their transport by the secretion system, their activity after secretion, and their different strategic use by the T6SS apparatuses in bacterial competition might play a role. To test these hypotheses, we complemented our data on effector occurrence in the strains of our dataset and the species' MRCA with what is known about the genomic organisation, transport, activity, and strategic use of the individual effectors (Fig. 10a–d and Supplementary Data 26). First, we analysed the effector occurrence by genomic organisation of the effector gene without or with an immunity gene and in- or outside of a T6SS genomic island with the structural T6SS genes *hcp*, *vgrG* and PAAR (genomic organisation type I–IV) (Fig. 10a). Effector genes without a T6SS genomic island (type I and II) were exclusively in the MRCA and core effectors, suggesting their ancient acquisition and little potential for intraspecific diversity. In contrast, accessory effectors are all effector genes in T6SS genomic islands (type III and IV), demonstrating a role of these elements in facilitating diversity, further supported by the earlier section on the mechanism of effector acquisition of such elements (e.g., T6SS genomic island with *vgrG14* and *rhsP2*). Interestingly, not all effectors of type III and IV in T6SS genomic islands show intraspecific diversity, suggesting that genomic organisation is not the only determining factor. Second, we tested effector occurrence according to an effector's covalent or non-covalent bond with structural T6SS proteins during secretion (Fig. 10b). We found that cargo effectors, which are defined as effector proteins with a non-covalent bond to structural T6SS components[11,63], were the most common amongst the core effectors. On the contrary, specialised effectors, which are defined as structural T6SS components with covalent bonds to effector domains and are one protein[11,63], were predominant among accessory effectors. Third, we thought that effectors might differ by the structural T6SS protein with which they are associated during secretion (Fig. 10b). Indeed, effectors that are loaded into the Hcp tube for T6SS firing were all exclusively in the species' MRCA and core effectors whereas

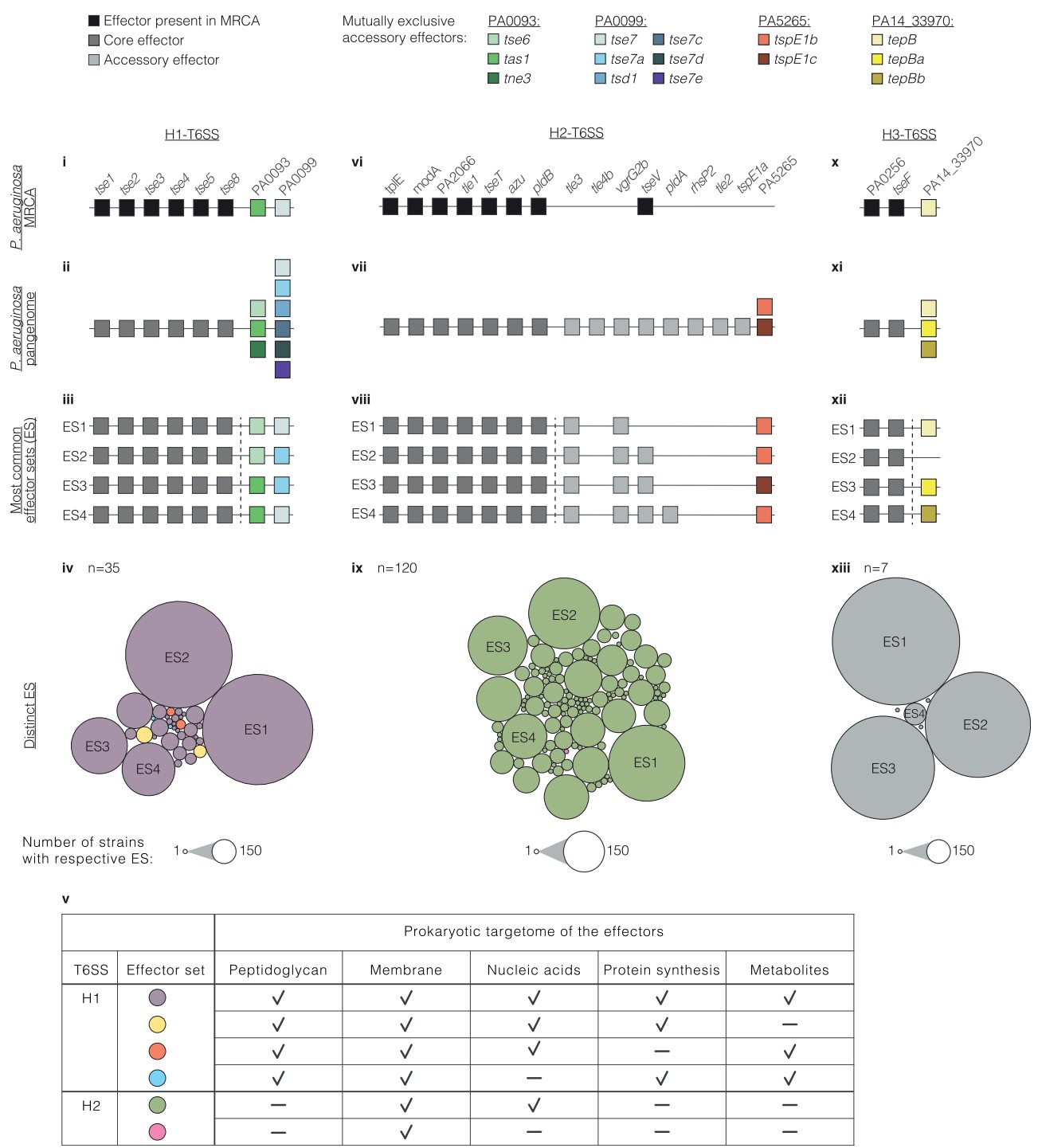

**Fig. 9 | Intraspecific diversity of T6SSs effector sets.** Graphical depictions of the ESs found in the MRCA (**i, vi, x**), the pangenome of H1-, H2-, and H3-T6SS effectors (**ii, vii, xi**) and the four most common ESs per T6SS (**iii, viii, xii**) in the dataset of *P. aeruginosa* strains (*n* = 1912) (list of all observed effector sets and their occurrence is shown in Supplementary Figs. 27–31). Core effectors are indicated in dark grey, and accessory effectors in light grey (with the exception of mutually exclusive accessory effectors in colours). Bubble charts indicate distinct ESs among genomes of the dataset (**iv, ix, xiii**). The size of the bubble indicates the occurrence of the respective ES. The bubbles are coloured according to the reported prokaryotic targetome (**v**) of the effectors in the ES (Supplementary Data 26).

effectors transported via the PAAR domain at the tip of the T6SS apparatus were predominantly accessory effectors. This conceptually aligns with the fact that only a single PAAR protein can be loaded to a T6SS, whilst many Hcp-loaded effectors can likely be present within the tube. Fourth, we tested effector occurrence according to effector activity toward a biotic target (e.g., cells) or an abiotic target (e.g., nutrients) after secretion (Fig. 10c). We found effectors with an abiotic

target and nutrient-acquiring activity were exclusively present in the species' MRCA and are core effectors. Fifth, we tested effector occurrence according to the strategic use of the T6SS in interbacterial competition (Fig. 10d). The H1-T6SS is considered a defensive weapon because it is inactive unless the bacterium is being attacked, and the H1-T6SS is activated in response to fight off the attacker[64,65]. In contrast the H2-T6SS is considered offensive because it is activated at high cell

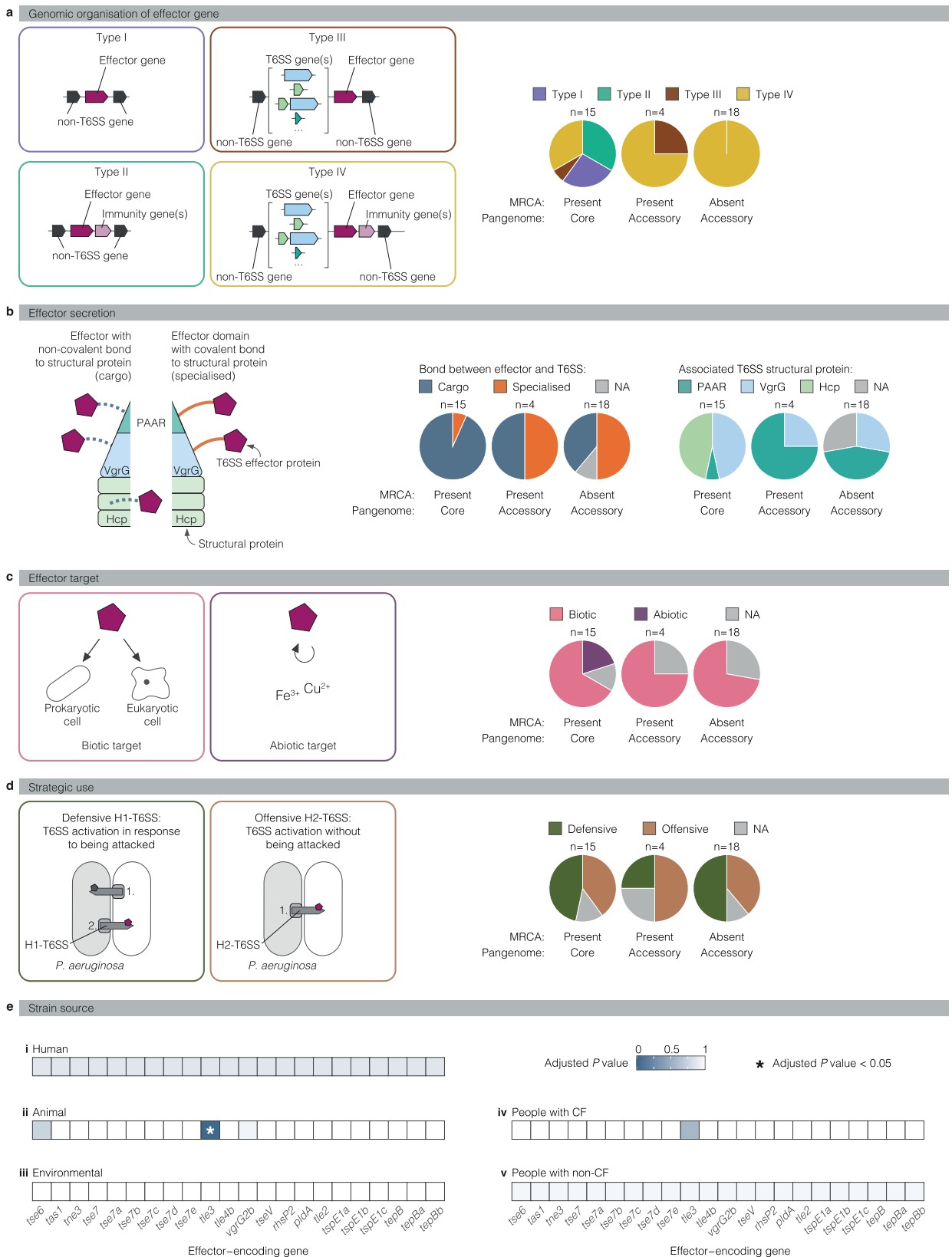

**Fig. 10 | Association of effector gene occurrence with effector characteristics and the strains' ecosystem. a–d** Analysis of shared characteristics amongst effectors with similar occurrence patterns. Models display the focus of the individual analyses as labelled in grey boxes. Pie charts indicate the number of effectors that share an indicated effector characteristic. Effectors are grouped based on their presence or absence in the MRCA and their classification as core or accessory T6SS effectors. **e** Association study shows effector gene *tle3* associated with an animal isolation source. Coloured boxes indicate the *P*-values of the association study. Significant associations were assessed with a hypergeometric test (*P*-value < 0.05, FDR = 0.05).

density and does not depend on an external cue to engage in killing other bacteria[66]. Both systems have similar numbers of core and accessory effectors, with some in the MRCA and others not. Interestingly, a smaller number of diverse effector sets was observed for defensive use by the H1-T6SS (Fig. 9iv), whilst those for offensive use by the H2-T6SS had a much higher number of diverse effector sets driven by the larger number of accessory effectors that were encoded at varying genomic loci and not as mutually exclusive effector genes at one locus (Fig. 9ix). In conclusion, we identified a non-random distribution of core and accessory effectors according to (i) genomic organisation of the effector genes (based on the first observation above), (ii) effector secretion by the T6SS (based on the second and third observation), (iii) effector activity towards a target (fourth observation), and (iv) strategic effector use in bacterial warfare (fifth observation).

Next, we assessed whether individual accessory effectors would be enriched by strains of a given ecosystem (Supplementary Figs. 37, 38). We tested for statistically significant associations between the presence of an effector gene and the isolation source of interest in a way that takes the phylogeny of the analysed strains into account. Remarkably, amongst the strains of animal, human, and environmental origin, we found *tle3* to be associated with animal origin (Hypergeometric test, FDR = 0.05, *P* < 0.05) (Fig. 10e i–iii and Supplementary Data 27). After correction for multiple testing, no statistically significant association was observed between an effector gene and the patient origin among the clinical isolates (Fig. 10e iv, v and Supplementary Data 28). A potential causal relationship between the presence of an effector gene and successful colonisation of the respective habitat awaits future experimental characterisation. However, the data presented here supports the idea that strains with a particular effector have an advantage in the colonisation of this environment.

## Discussion

Here, we report the distribution of T6SS apparatus and effector genes in a global population of *P. aeruginosa*. Our results provide an understanding of the T6SS within a bacterial species that builds upon what is known about the secretion system from molecular microbiology, biochemistry, and structural biology and expands the knowledge of the intraspecific diversity of *P. aeruginosa* T6SSs. Current studies estimate that the core genome of this species comprises a small percentage of the pan-genome[67–69]. Strikingly, our data shows that within this small pool of core genes (12% of the pan-genome of our dataset, Supplementary Data 3, 29) are those encoding for three T6SS apparatuses and fifteen of their effectors. We therefore consider T6SS-mediated interactions with other cells and nutrient uptake as universal traits of bacteria within this species and central to their lifestyle. A fourth T6SS apparatus and numerous accessory effector genes for the other systems are only found in a fraction of strains, which is observed for T6SSs of several other bacterial species[70–73].

Here, we define the core and accessory effectors of the T6SS in *P. aeruginosa*. This concept has already been established for other secretion systems[74–79] and provides a conceptual framework for T6SS effector diversity beyond *P. aeruginosa*. Core effectors are highly prevalent and conserved across strains of the species. In sharp contrast, accessory effectors differ between strains, with some being mutually exclusive. It is tempting to speculate that core effectors mediate interactions with competing cells that are common to diverse environments of *P. aeruginosa* strains, whereas accessory effectors mediate a selective advantage in a strain's local environment or against niche-specific competitors. Indeed, we observed conservation across *P. aeruginosa* genomes for effector genes that encode proteins with abiotic targets involved in metal ion acquisition. The expression and deployment of such effectors would have a clear benefit in low ion conditions but also likely do not trigger a potential retaliatory cost – in contrast to antibacterial effectors, which may promote bacterial

antagonism and counterattack[64,80,81]. Moreover, defining core and accessory effector repertoires has recently been shown to help decipher T6SS machines' roles in targeting eukaryotic or prokaryotic cells as their primary functions[82,83].

Considering that multiple different effectors are translocated simultaneously by a single T6SS apparatus, synergy between these effectors is a long-standing topic in the field. We found effectors known to act synergistically[33] to be core effectors. Synergy might also arise between accessory and core effectors, for example, in their abilities to rapidly lyse competing bacteria[84]. In our analysis, we observed core effectors (Tse1, Tse3, Tle1, PldB) and accessory effectors (Tle3, VgrG2b, PldA) that are predicted to induce lytic cell death. We find an indication of synergy in our co-occurrence analysis amongst H2-T6SS accessory effectors.

Our data suggests horizontal gene transfer as one evolutionary mechanism for the emergence of intraspecific diversity in T6SS effector genes during the genetic diversification of the species with three T6SS systems and 15 core effectors key to the lifestyle of *P. aeruginosa*. We show differences in effectors within phylogroups and evidence of horizontal transfers between them. We show that the effector sets of the *P. aeruginosa* and *P. paraeruginosa* MRCA differ, which supports the idea that the T6SS system and its effectors likely facilitate competition between these species and may contribute to the species boundary. We also show evidence of effector gene transfer across species boundaries, and a recent report on the transfer of an effector gene between *Vibrio cholerae* strains and the closely related *Vibrio anguillarum*[85] shows this occurs more broadly in Gram-negative bacteria. The methods used in our study could be applied to other T6SS-harbouring species to reveal their T6SS effector inheritance and evolutionary history, advancing the overall paradigm of T6SS evolution and roles. Existing experimental work further demonstrated the acquisition of effector genes upon natural transformation and by site-specific recombination[61,86,87]. What is particularly striking to us are the T6SS genomic islands with *vgrG* genes, which likely arose by duplication as observed for the *vgrG1abc* or *vgrG2a/2b* islands, and the subsequent much more frequent swapping of effector genes within these T6SS islands by homology-facilitated illegitimate recombination. With a limit to the number of effectors that can be secreted with one shot of the T6SS, there may not be an advantage from simply accumulating effector genes. The exchange of effector genes directly embedded in a regulatory network and in a manner that guarantees secretion with existing structural proteins encoded in a specific T6SS genomic island seems a very elegant mechanism that could allow rapid adaptation to the ebb and flow of diverse bacterial competitors in the immediate environment. In addition, the acquisition of a new rare effector-encoding gene could allow them to displace their ancestors due to the negative frequency-dependent selective advantage.

Finally, we found an extremely large biodiversity in T6SS effector sets in a natural population of *P. aeruginosa*. Indeed, laboratory strains neither encode all T6SSs of the species nor do they represent the most prevalent combined effector set. The population genetic approach used here to study this unique secretion system and its effector genes brings clarity to the discovery of a large variety of effectors and their importance, variation, and roles in the lifestyle, competitive interactions, host manipulation and evolution of *P. aeruginosa* as an opportunistic pathogen present in diverse abiotic and biotic environments.

## Methods

### Accession of genomes and assessment of genome quality
We downloaded 1960 assembled *P. aeruginosa* genome sequences from NCBI using Entrez direct[88] on July 15th, 2022. We chose this dataset because the genomes cover the phylogenetic diversity of the species, belong to strains that were isolated from every continent, and include strains of clinical and environmental origin (summarised in Supplementary Data 4). The genomes had previously been used in a

large-scale bioinformatics study by Botelho et al., who performed initial quality control and reported on a Mash distance between the genomes of 0.0001 to 0.05[69]. Out of the 1960 genomes, 239 existed as complete genomes. We performed additional analyses to assess genome quality, completeness, and test for contamination. QUAST[89,90] (version 5.2.0) was used to evaluate the quality of the genome assemblies. N50 values were between 111,311 and 7,564,383 with a mean of 1,624,041, which is over 1 Mb and generally considered good. Genomes that consisted of more than one contig had L90 values between 2 and 86, which is below the threshold of 100 and an indication for little fragmentation[91]. To assess the completeness of the assembled genomes, we tested for the presence of universal single-copy orthologs using BUSCO[92,93] (version 5.7.0). The lineage dataset pseudomonadales_odb10 was used, which includes 782 universal single-copy orthologs for the order *Pseudomonadales*. Our genomes had BUSCO scores between 97.5% and 100%, which is considered very close to complete. To test for contamination, we analysed the genome's GC% content using QUAST. The GC% content of non-overlapping 100 bp windows followed a Gaussian distribution (Supplementary Fig. 39), which is an indication for non-contaminated assemblies[89]. In sum, we consider the 1960 genomes in our dataset of sufficient quality for our analysis, given the quality of assembly, level of completeness, and no sign of contamination. The accession codes and additional information on the genomes (source of isolation, sequence type (MLST), quality of assembly, level of completeness) are provided in Supplementary Data 2.

### Identification and classification of the H4-T6SS apparatus gene cluster

A local blastn search was performed to identify genes in the dataset of *P. aeruginosa* genomes with homology to structural genes of the known three T6SS apparatus gene clusters. The nucleotide sequences of *tssB1* (PA0083), *tssB2* (PA1657), and *tssB3* (PA2365) of the reference strain PAO1 (NC_002516) were used as a query, which encode the TssB protein that forms the contractile outer sheath of the secretion apparatus, is essential for T6SS function, is well understood, and is widely used in experimental studies as hallmark of the T6SS[94]. The search revealed hits with an E-value below $1 \times 10^{-5}$ in a few genomes, including *tssB4* of strain LYSZa7, which has been reported independently by Ren et al.[27]. The genes encoded adjacent to the hits were inspected with Geneious (version 2019.2.3) and annotated manually based on their similarity to T6SS genes in the known apparatus gene clusters. The completeness of the gene cluster was assessed by testing for the presence of all genes that encode essential proteins for a functional secretion apparatus. The synteny of genes in the apparatus gene clusters was determined and used as a basis for the classification as an 'H4-T6SS' separate from an H1- to H3-T6SS, building onto the existing distinction between H1-, H2-, and H3-T6SS apparatus gene clusters that differ between each other in their synteny. To classify the H4 apparatus gene cluster according to an established classification system of the T6SS field that is based on very diverse T6SSs from a wide range of bacterial species, we used the SecReT6 database[95] and the amino acid sequence of TssB4 of strain 60.1 as a query. To assess the similarity between the H4-T6SS apparatus gene clusters in *P. aeruginosa* genomes of the dataset, the nucleotide sequences of the clusters were extracted, aligned with MUSCLE[96] (version v3.8.31), and visualised with Geneious[97] (version 2019.2.3).

### Occurrence analysis of T6SS apparatus genes

A local blastn search was performed to measure the occurrence of the apparatus gene clusters directly in the dataset of assembled genomes without annotating the genomes. As queries, we used the individual genes of the three previously reported T6SS apparatus gene clusters of the reference strain PAO1 (H1-T6SS: PA0074-91, H2-T6SS: PA1656-71, H3-T6SS: PA2359-2373, PAO1 accession number: NC_002516)[98,99], and

the H4-T6SS apparatus gene cluster of the strains LYSZa7 (GCF_016584725.1) and 60.1 (GCF_900148005.1). As a database, we used the dataset of assembled genomes (Supplementary Data 2). The search was performed with a nucleotide-nucleotide blast (version 2.10.1 + ) using default settings and adjusting -max_target_seqs to 5000. We chose the output format 10 with the following output parameters: qseqid, qlen, qstart, qend, sacc, slen, sstart, send, qseq, sseq, length, nident, pident, bitscore, e-value. Based on these parameters, we additionally calculated a nucleotide identity score of the blast hits to the query as described in detail by Rohwer et al.[100]. In short, blast returned the percentage of identical matches ('pident') over the length of the alignment between the query and the subject, which did not take the total length of the query sequence into account. To account for the length of the query sequence, we calculated nucleotide identities like Rohwer et al. with the following equation using the parameters returned by the blastn analysis: nucleotide identity = (pident*length)/(qlen + (length-(qend-qstart + 1))). Sequences with an identity score of 95% or higher were considered to be a hit. Genomes with hits for at least 12 apparatus genes were considered positive for the respective T6SS. All genomes in which we got blastn hits with an identity score below 95%, or no blastn hit at all, were additionally analysed with MacSyFinder[101] (version 2.1.1) (--models TXSScan all) to independently test for the presence of a T6SS apparatus gene cluster. Among the genomes with no T6SS hit, exemplary genomes were additionally inspected manually using the progressiveMAUVE algorithm in Geneious (version 2019.2.3) to ensure the genomic locus at which an apparatus gene cluster is expected is located within a contig, and the absence of the gene cluster is not an artefact introduced by the assembly state of the genomes. To separate between genes of the H1-, H2-, H3- and H4-T6SS, we performed a reciprocal best hit analysis using blastn. The percentage occurrence of the apparatus genes was calculated based on the number of genomes with T6SS apparatus genes and the total number of genomes analysed. The co-localisation of apparatus genes in a cluster was checked separately by computing multiple sequence alignments of the entire gene cluster and confirmed (H1-T6SS: $n = 1605$, H2-T6SS: $n = 1949$, H3-T6SS: $n = 1883$, H4-T6SS: $n = 19$; all genomes with apparatus genes on one contig).

### Occurrence analysis of T6SS effector genes

To measure the occurrence of effector genes in the dataset of assembled genomes, we performed a search with blastn. Nucleotide sequences encoding for known *P. aeruginosa* T6SS effectors were extracted from annotated reference strains PAO1 (NC_002516), PA14 (NC_008463.1), PAK (NZ_CP020659.1), and from clinical strains PALA36 (CP110348) and PALA52 (CP110347), and used as queries (Supplementary Data 1). In the case of *vgrG2b*, only nucleotides 2271 to 3060 were used as queries because most of the protein encoded by this gene contributes to the structure of the T6SS and the enzymatically active domain with effector function is comparatively small. The blastn search and calculation of the nucleotide identity score were performed as described for the occurrence analysis of T6SS apparatus genes. Genomes with a hit above 80% nucleotide identity were considered positive for the respective effector gene. Among the genomes that were negative, individual genomes were inspected manually, the genomic locus identified at which an effector gene is located in other strains, the nucleotide sequence spanning one gene upstream and downstream of this locus extracted, and used as a query for a new blastn search. Genomes with a hit above 80% identity for this locus without an effector gene were then considered negative for the respective effector and ensured that the absence of an effector gene at this locus is not an artefact introduced by the genomes' assembly state. All genomes that had neither a positive nor a negative hit for an effector gene in the blastn search were further inspected manually with Geneious[97] (version 2019.2.3). The percentage occurrence was calculated based on the number of genomes with a particular effector

gene and the total number of genomes analysed. In the pan-genomic analysis, the established threshold of at least 95% occurrence for genes of the soft-core genome[52] was applied to consider an effector gene a core effector or an accessory effector.

## Phylogenetic analysis of genomes

All species trees were built on a core genome alignment in multiple steps. First, we annotated the genomes with Prokka[102] (version 1.14.5). Second, we performed a pangenome analysis using Panaroo[103] (version 1.3.2) with default settings and mode 'moderate' on the annotated genomes. The genes of the core genome (threshold of 99% occurrence to be considered a core gene lists of marker genes for each tree are provided in Supplementary Data 30–32) were used by Panaroo to build a concatenated core genome alignment that is used as the basis for the tree as described towards the end of this section. The best-fit model for the phylogenetic tree was chosen with ModelFinder[104]. Due to computational limitations, the model was chosen not based on the core genome alignment but based on separate alignments of individual genes of the core genome. For each core gene, sequences of ten randomly chosen genomes were used to build a multiple sequence alignment using MUSCLE[96] (version v3.8.31) and the best-fit model for each of those alignments was determined based on the Bayesian Information Criterion (BIC) with ModelFinder[104]. We enumerated how often which model was chosen (Supplementary Data 33) and the model that was most often considered the best fit was used for further analysis and is indicated in the figure legends for each tree. The maximum-likelihood phylogenetic tree was then inferred on the core genome alignment (output file 'core_gene_alignment_filtered.aln' from Panaroo) using IQ-tree[105,106] (version 1.6.12) with the respective best-fit model and 1000 ultrafast bootstrap replicates. The trees were midpoint or outgroup rooted, which is indicated in the figure legends, and plotted using the R package ggtree[107]. Circular heatmaps indicating the presence of T6SS apparatus gene clusters or effector genes were added with the function gheatmap() of the R package ggtree.

## Phylogenetic analysis of individual effector genes and apparatus gene clusters

All gene trees were built in multiple steps. First, we performed a multiple sequence alignment based on individual effector or apparatus genes using MUSCLE[96] (version v3.8.31). The alignments of apparatus genes were then concatenated. Second, the best-fit model was chosen individually for each tree based on the Bayesian Information Criterion (BIC) using ModelFinder[104] and is reported in the legends of the respective figures. The maximum-likelihood tree was then inferred with IQ-tree[105,106] (version 1.6.12) based on 1000 ultrafast bootstrap replicates. The trees were either midpoint rooted or unrooted, as also indicated in the respective figure legends, and plotted using the R package ggtree[107].

## Co-phylogenetic analysis of species tree and effector gene tree

The co-phylogenetic plots were generated and plotted with R[108] (version 4.2.2) using the cophylo() function of the R package phytools[109,110]. These plots are directly based on the species tree and gene tree, which were built separately as described in the respective sections of the methods.

## Reconstruction of ancestral states, measurement of gene gain and loss, and calculation of transition state diagrams

To reconstruct the presence or absence of effector genes in ancestral strains of the species, we performed stochastic character mapping. Therefore, the R package phytools[109,110] was used as described by Revell[110]. Each effector gene was mapped onto each node of the species tree. This phylogenetic tree was built as described in the separate section of the methods based on 1912 *P. aeruginosa* genomes (all genomes in the dataset with all three apparatus genes clusters of the

H1-, H2-, and H3-T6SS) and an outgroup of five genomes from the closely related species *P. paraeruginosa* and seven genomes of distinct *Pseudomonas* species (*P. delhiensis, P. humi, P. jinjuensi, P. knackmussii, P. multiresinivorans, P. nitroreducens, P. panipatensis*) that had also been used as an outgroup in an existing study[53] (accession codes of all genomes used are listed in Supplementary Data 15). More specifically, we used effector presence and absence as a binary trait and fitted four character transition models (ER: equal rates model, ARD: all-rates-different model, and two different irreversible trait evolution models) using the fitMk() function. The fitted models were then compared, and their Akaike Information Criterion (AIC) was calculated using the generic anova() function. Stochastic character maps were sampled with the simmap() function (nsim = 1000) using all four fitted models weighted by their relative frequency equal to the evidence supporting each model (Akaike model weights)[110]. We then calculated the posterior probabilities for all effector genes at each ancestral node with the summary() function. Effector gene presence or absence was inferred at each node from the posterior mode of the point estimates, i.e., an effector gene was considered present at nodes at which the posterior probability was >50%. We report the inferred presence of effector genes at the node that corresponds to the *P. aeruginosa* MRCA with a graphical depiction.

To quantify effector gene gain and loss events during the diversification of the species, we used the data of the stochastic mapping and analysed effector gene presence and absence across consecutive nodes. This analysis was performed merely based on the presence or absence of an effector gene in a genome regardless of where in the genome the effector gene is encoded. A gain event was inferred when the point estimates changed from 'absence' to 'presence' between two consecutive nodes, and a loss event when the point estimates changed from 'presence' to 'absence' between two consecutive nodes. To estimate the total number of gene gain and loss, the density() function was used. The results were plotted in R using ggplot2[111].

A second analysis was performed with a focus on the genomic loci and all effectors encoded at a respective locus. Therefore, we used information on the presence or absence of effector genes encoded at the same genomic locus as our trait information. Stochastic mapping was performed as described above. The density() function was used to estimate the overall number of character transitions, i.e., how often one effector gene is gained, lost, or replaced by another effector gene at the same genomic locus. For visualisation, state transition diagrams were generated using ggplot2[111] with the extension ggraph[112] and data of inferred gain or loss events of effector genes are visually marked as stars in separately generated phylogenetic trees.

## Detection of recombination breakpoints

We tested for recombination events by performing the following analysis in multiple steps. First, we randomly selected *P. aeruginosa* genomes of our dataset with either variant of the effector gene at the genomic locus of interest. Second, nucleotide sequences were extracted that span the effector gene and neighbouring genes in either direction. Third, we calculated a multiple sequence alignment using MUSCLE[96] (version v3.8.31). To identify recombination events, we used four different algorithms (GENECONV, RDP, MAXCHI, CHIMAERA) provided by RDP5[113]. Putative recombination events detected by at least three of the four algorithms[114] were included in the remaining steps of the analysis. Next, recombination events were manually refined following the instructions of the RDP5 manual. More specifically, recombination events were confirmed or denied based on the pairwise identity plot of the RDP algorithm and the -log(chi2 *P*-value) of MAXCHI and CHIMAERA. Recombinant sequences were determined by comparing the position of the recombinant sequence in two phylogenetic trees. To identify recombination hotspots, the number of breakpoints was calculated per 200 nucleotides and a test was performed to assess whether this number exceeds the number of

breakpoints expected by chance. The results were plotted in a break-point distribution plot using RDP5.

## Search for homologues of *P. aeruginosa* TssB4 in other species

To identify homologues of *P. aeruginosa* TssB4 in other species, we used blastp provided by the blastp suite of NCBI. The protein sequences of TssB4 were used as a query. One reason for this choice is that the nucleotide sequences did not reveal strong hits in a blastn search. *Pseudomonas aeruginosa* (taxid: 287) was excluded, and the search was performed on the non-redundant protein sequences. BLAST hits with a query coverage of above 95 percent (a measurement of BLAST to assess the length of the alignment between query and subject relative to the query length), and identity of above 70 percent (a measurement of BLAST to assess the percentage identity along the alignment of query and subject) was considered homologues. Up to five top hits were chosen for further analysis. The genes encoding these TssB4 protein homologues and the genomes that harbour these genes were included in the co-phylogenies.

## Search for homologues of *P. aeruginosa* T6SS effector genes in the genomes of other species

To identify homologous sequences of *P. aeruginosa* T6SS effector genes in other species, we used megablast provided by the blastn suite of NCBI. The nucleotide sequences of *P. aeruginosa* effector genes were used as a query, and the search was performed on non-redundant standard databases of NCBI, excluding *Pseudomonas aeruginosa* (taxid: 287). Among the hits provided by BLAST, genes with above 80 percent query coverage (a measurement of BLAST to assess the length of the alignment between query and subject relative to the query length) and above 70 percent identity (a measurement of BLAST to assess the percentage identity along the alignment of query and subject) were considered homologues and top hits were included in the co-phylogenies. For effector genes for which we did not get hits, we used the amino acid sequences of the encoded effector protein as a query. This search was performed with blastp of NCBI on the database of non-redundant protein sequences excluding *Pseudomonas aeruginosa* (taxid: 287). Hits with a percent query coverage of above 85% and percent identity of above 30% were considered homologues, and up to the top three hits were included in the co-phylogenies.

## Analysis of genomic loci with T6SS genes across strains

To test whether the T6SS apparatus or effector genes are located at the same genomic locus across acquisition events, we investigated the position of the gene(s) of interest relative to adjacent genes of the core genome. First, we took an exemplary genome that contained the gene(s) of interest and identified the closest non-T6SS genes of the core genome upstream and downstream. These genes of the core genome are expected to remain stable in the genome and serve as an anchor in our analysis. The information on the core genes was retrieved from the pan-genome analysis. Next, we used customised scripts to extract the nucleotide sequence spanning the first non-T6SS core gene upstream and downstream of the gene(s) of interest and used this sequence as a query in a blastn search as described in the section on the occurrence analysis against all strains that encoded the gene(s) of interest. The percentage identity of the hit to the query was calculated as described in the section on the occurrence analysis. Hits with an identity score of >95% were considered positive for the gene(s) of interest being adjacent to the same non-T6SS genes of the core genome. Strains with non-conclusive hits were analysed manually in Geneious[97] (version 2019.2.3).

## Analysis of effector sets

The combination of effector genes in the *P. aeruginosa* genomes of the dataset was determined by analysing the presence and absence of effector genes by genome. This analysis made use of the data on

effector gene presence and absence per genome that was generated as part of the blastn search described in the methods section on the occurrence analysis of the effector genes. More specifically, the data on the presence and absence of effector genes was organised by genome and analysed for the combination of (i) effector genes associated with the H1-T6SS, H2-T6SS, or H3-T6SS, and (ii) all effector genes in the genome regardless of the T6SS they are associated with. The number of distinct effector sets was determined by grouping all genomes with the same effector set and enumerating the number of distinct sets. The number of genomes with an effector set was determined by enumerating the number of genomes with each distinct set. The results were processed with R using the package packcircles and its function circleProgressiveLayout() and visualised as bubble charts using ggplot2[111]. All effector sets of genomes of the dataset and effector sets of clinically relevant common sequence types (MLST) are reported as graphical depictions. The number of theoretically possible effector sets was calculated by multiplying the number of observed states at a genomic locus (e.g., two states for a locus with or without an effector gene across genomes of the dataset) for all loci of an effector set.

The rarefaction curve analysis was performed with a customised script in R[108]. The data on the effector sets of the 1912 *P. aeruginosa* genomes with all three apparatus gene clusters of the H1-, H2-, and H3-T6SS was sampled 10,000 times and for each sampling, the number of distinct effector sets per sampled genomes was recorded. Results were plotted as a rarefaction curve in R using ggplot2[111].

The co-occurrence of any two separate accessory effector genes in the same effector set across the species tree was tested with the R package PhyloCorrelate[115]. All pairwise combinations of accessory effector genes were tested while taking the topology of the species tree into account. The binary data on the presence or absence of effector genes (see methods section on occurrence analysis of effector genes for details) and the species tree (see methods section on phylogenetic analysis of genomes) were used as input. The runs-adjusted hypergeometric *P*-value was calculated using the doParCalc() function. *P*-values were corrected for the false discovery rate (FDR) with the p.adjust() function in R using the Benjamini-Hochberg method. An association with an adjusted *P*-value of < 0.05 was considered statistically significant.

## Association study between effector occurrence and effector characteristics

For this study, the effector genes' occurrence was analysed by (i) the genomic organisation of effector genes as described in Fig. 4, (ii) the mechanism of secretion of the effector protein encoded by the effector gene and (iii) the strategic use of the T6SS that the effector is associated with. This analysis made use of the classification of *P. aeruginosa* effector genes as core or accessory as described in the methods section on the occurrence analysis, and the data on the effector gene presence or absence in the *P. aeruginosa* MRCA as described in the section on the reconstruction of ancestral states. Categorical data on the effector genes' genomic organisation was generated by manual inspection of the genes adjacent to T6SS effector genes in the reference strains PAO1 (NC_002516), PA14 (NC_008463.1), PAK (NZ_CP020659.1), and clinical strains PALA36 (CP110348) and PALA52 (CP110347). An effector gene without or with an immunity gene surrounded by non-T6SS genes is considered type I and type II, respectively. Effector genes surrounded by other T6SS genes that encode structural T6SS proteins like VgrG, Hcp, or PAAR, but were not found in a pair with an immunity protein-encoding gene were considered type III. Effector genes surrounded by other T6SS genes and found in a pair with an immunity protein-encoding gene were considered type IV. Additional categorical data was generated by revisiting published experimental data on the mechanism of secretion of the *P. aeruginosa* T6SS effector proteins and the different strategic uses of

the *P. aeruginosa* T6SSs in bacterial warfare. A summary of the data and references to studies that the data is based on are listed in Supplementary Data 1 and 26. All data was entered into a master file, the effectors were grouped by occurrence, and the relative frequency of the categorical variables was assessed for each category. The results were visualised as pie charts in R[108] using ggplot2[111].

## Association study between a strain's effector genes and isolation source

The association between a strains' effector gene and isolation source was tested with the R package PhyloCorrelate[115]. Importantly, this package corrects for the strain's phylogeny when testing for co-occurrences between two traits and thereby avoids an association being considered statistically significant based on the mere presence of two traits in the same lineage. Although originally intended for the study of co-occurrences between two genes, we used the tool for the study of co-occurrence between a gene and a strain's isolation source, which we consider valid because of the input of binary data in either case and a similar biological meaning of an association between two proteins that might affect each other within a cell, which is the originally intended scenario, and a protein that is secreted by the T6SS and acts on the strains' environment, which is the scenario of our study. As input, we used (i) binary data on effector gene presence and absence (e.g., strain X has T6SS effector gene *pldA*: yes/no) that was generated in the above described blastn search (see methods section on occurrence analysis of the effector genes), (ii) binary data on the strains' isolation source (e.g., strain X was isolated from human: yes/no) that was retrieved from NCBI when available and is listed in Supplementary Data 2, iii) the species tree (see methods section on phylogenetic analysis of genomes). For each combination of effector and isolation source, we used the doParCalc() function to calculate the runs-adjusted hypergeometric *P*-value. The data was plotted in R[108] using ggplot2[111], and a *P*-value of < 0.05 was considered statistically significant.

## Calculation of GC% content

The GC% content of the whole genomes was calculated by QUAST[89,90]. The GC% for T6SS apparatus genes of the H1-, H2-, H3-, and H4-T6SS were calculated with a customised Python script that extracts the 12 apparatus genes (*tssA, tssB, tssC, tssE, tssF, tssG, tssJ, tssK, tssL, tssM, clpV,* and *hcp*) from the genome, concatenates them, and calculates the GC%.

## Graphical depictions

Illustrations in this manuscript were created with Adobe Illustrator (version 28.0).

## Reporting summary

Further information on research design is available in the Nature Portfolio Reporting Summary linked to this article.

# Data availability

Accession codes of analysed genomes are provided in Supplementary Data 2, 13 and 14.

# Code availability

Scripts used for analyses are available at GitHub (https://github.com/UnterwegerLab/2024_Habich_et_al).

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

## Acknowledgements

We thank the Unterweger group for their discussions and comments on the manuscript. We thank Tal Dagan and John Baines for the discussions and sharing resources. Work in the Unterweger group is supported by the German Federal Ministry for Education and Research (grant 01KI2020), the Deutsche Forschungsgemeinschaft (RU5042, EXC2167, CRC1182), the German Cystic Fibrosis Association, and the Daimler and Benz Foundation (grant 32-10/18). A.H. and V.C.V. receive support from the International Max Planck Research School for Evolutionary Biology. Work in the L.P.A. laboratory is supported by the Academy of Medical Sciences/the Wellcome Trust/the Government Department of Business, Energy and Industrial Strategy/the British Heart Foundation/Diabetes UK Springboard Award [SBF006\1161] and by a Biotechnology and Biological Sciences Research Council grant BB/Y00048X/1. L.P.A. is also supported by a NIHR Imperial Biomedical Research Centre (BRC) Grant as well as funds from the National Heart and Lung Institute at Imperial College London. L.A.R. PhD studentship is funded by the AMS and the NHLI.

## Author contributions

A.H. and D.U. conceived the project. A.H., V.C.V., and L.A.R. performed the bioinformatics. A.H., V.C.V., L.A.R., L.P.A., and D.U. analysed the data. D.U. and A.H. wrote the first draft of the manuscript, which was revised by all authors. D.U. and L.P.A. supervised the study.

## Funding

## Competing interests

The authors declare no competing interests.
