## [Transparent Peer Review file · Nature Communications]

Distribution of the four type VI secretion systems in *Pseudomonas aeruginosa* and classification of their core and accessory effectors

Corresponding Author: Dr Daniel Unterweger

Version 0:

Reviewer comments:

Reviewer #3

(Remarks to the Author)

I thank the authors for this revised version of the manuscript. Unfortunately, to me the main criticisms made by reviewer #2 still hold as they were not sufficiently addressed. Here follow more details.

Issues with evolutionary interpretation.

Despite the authors' efforts to increase the discussion of the evolutionary aspect in relation to the observed diversity –in particular with the addition of the MRCA and effectors' dynamics part, I still have the feelings that this aspect is insufficiently covered while being indissociable of the work on the diversity of the effectors. Changing the title of the paper is not enough to alleviate the fact that it seems missing from the study. See also my comments below. In general, the mechanisms at the source of the origins and diversity of i) the different T6SS loci and ii) the different effectors could have been investigated in deeper details. On several occasions, potential mechanisms are invoked, but not further investigated. This includes in particular homologous recombination and lateral gene transfers. For instance, accompanying the gain/loss scenarios of T6SS effectors with phylogenetic analyses could have helped settling for vertical versus horizontal transfers (or duplications). Further, a study of the genomic locations and genomic context of the T6SS loci and effectors, could have helped to address the mechanisms of transfers. For instance, are they all located at the same loci? Are there traces of mobile genetic elements around? Etc... See here more specific comments:

- Regarding the effectors' evolutionary history, reconstruction of individual gene phylogenies would have been needed to complement and support the gain/loss scenario. Gene trees should have been presented in a detailed manner. For instance, if only one acquisition event is inferred, it is expected that the gene history globally follows the species history after acquisition (and harbour the same topology). It does not seem enough to look for potential homologs in other bacteria as the authors did for some effectors. And even here, details are missing on how was led this search: search performed how? in which bacterial genomes? (see also point on Methods). As the authors say themselves, it "remains to be analysed in depth."
- More evolutionary analyses are discussed in the text (lines 277-280) for some of the effectors with a more scattered distribution. But only two were presented, in Figure 6D-E. What happened with the other effectors?

- The authors used PhyloCorrelate in order to test for associations with traits (isolation source). However, PhyloCorrelate was designed to assess gene co-occurrence in a phylogenetic context. Could the authors elaborate on why they think that this method is well-suited to test the association between effector and traits?

How and why H4-T6SS could be singular compared to H1/2/3.

When reading the main text, it is still unclear what are the shared characteristics of the H4 T6SS and what makes them different from the other three loci. Especially when the authors name them H4a and H4b, whenever the phylogenetic tree seems to indicate there might be even more subclades. In Sup Fig 2, why not show the two loci H4a and H4b? It is actually shown later in Sup Fig 6 and it appears that the loci are almost identical. If the hypothesis of lateral gene transfers is invoked, why not pursue larger phylogenetic analyses to check the hypothesis of multiple acquisitions? (see also my general comment above). Maybe these independent acquisitions could justify to give H4 different names for the different sub-clades.

Database, dataset assembly and tree methods information are missing

There are still many missing pieces of information about the materials and methods used in this study. Here is a non-exhaustive list:

It is still unclear what is the nature and quality of the genome datasets used.

- Line 107: please specify which dataset of *P. aeruginosa*. Also, from Methods and Sup mat, the genomes used seem not to be complete genomes. The authors should provide statistics on genome assembly state. Can they be considered of “high” quality? What is the level of completeness? Contamination? How does this affect T6SS annotation? None of this is discussed or detailed.
- In link, line 195: what is the “global *P. aeruginosa* population”? Please explain dataset upon 1st mention in Results.

More details are also required for the methods:

- Details are missing on how were the T6SS annotated. Blastn is invoked, but which parameters? And what is the “query coverage”? (line 445). What means “taking coverage into account”? Line 469: what “testing for the direct neighbourhood of genes” means?
- Line 476-477: “To separate effector proteins, the nucleotide sequences of effector genes that were encoded at the same location across genomes were translated and aligned using the Geneious alignment” – how was “the same location” defined?
- Line 91: and on Figure 1C concatenation of which proteins?
- Why use the GTR model for phylogenetic reconstruction, and not try to select the best evolutionary model (e.g. IQ-Tree does that automatically)?
- How were the candidate donors found for H4-T6SS? Was it from a systematic search in all available complete genomes? Please explain the dataset used and general approach employed.
- Why choose TssB to compute the phylogenies (Figure 1C), and not a conserved, longer protein? Also, please remind the proteins’ role in T6SS upon 1st mention.
- Line 179: one cannot measure such low sequence identity, this makes no sense as sequences are too divergent to be aligned. Please clarify.
- Line 223: the general principles of the approach used to infer ancestral states is not mentioned in Results, neither are provided the details of which model among the four tested was finally used to produce the results.

Questionable usefulness of the statistical analysis of combinations of effectors

- I still do not really see the point of the effectors’ combination analysis, especially when means for the diversification of these combinations are not discussed.
- The occurrence analysis does not take phylogenetic inertia into account (as the authors tell themselves), whenever it is responsible of most of the co-occurrences observed – as further questioned by the very few positive associations found. I question the relevance of keeping it in the text.

Some concepts are clumsily introduced and convoluted.

- Lines 157-160. I did not understand the concept of “mutually exclusive effectors” when reading the manuscript at first. Could the authors please clarify? I only could understand because I read the authors’ response to reviewer #2. Also, no evolutionary mechanism is discussed here. It kind of falls flat, which is a pity as the sources of this diversity would be very interesting to unravel (see my above comment).

Other comments

- Why not explicitly state in which strain(s) the H4 cluster was first discovered? From the text, it seems that it has been missed/overlooked from many studies of *Pseudomonas aeruginosa*’s genomes. Yet it is found in only a few *P. aeruginosa* strains – which explains why it has been so far overlooked.
- Figure 1: coloring gene families according to family of homologs would help to visualize genomic organization

commonalities/differences between the loci.
- What are cargo and specialized effectors? Please define.

Version 1:

Reviewer comments:

Reviewer #3

(Remarks to the Author)

I would like to thank the authors for the considerable amount of work and efforts they put in this round of revision. I am happy with the replies to my comments and my concerns have been addressed. I believe this article now provides a lot of insights on the mechanisms of T6SS' and effector sets' diversification in *Pseudomonas aeruginosa*. The study addresses understudied, long-standing issues in the biology of secretion, in particular that of the T6SS. The unraveled diversity of the evolutionary mechanisms involved here should be very interesting to anyone interested in bacterial evolution. Moreover, the methods are now provided with a level of details sufficient for the reproducibility of the results and the discussion of the possible limits of the methodology. I still have some minor comments, as they are many new elements to account for.

In general, support values for phylogenetic trees are missing, clouding the interpretation of the significance of branchings. These should be added. It could be just an indication of the branches with a support e.g. >90% or 95%. This is particularly missing for Fig 3e.

The statistical tests results should be provided whenever they are mentioned (e.g. name or type of test, and p-value).

Fig.1. Please indicate what the colors correspond to? Or a shared color? Would it be possible to add on the figure the strains with 2 loci H4-T6SS?

Fig. 3. The font size is small.

L. 212-215. I don't understand here, as first the number of effectors of strain PAO1 is mentioned, and then it is said that "Effectors of H4-T6SS were omitted here". But is not PAO1 missing the H4-T6SS locus? As shown on Fig 2d? Please clarify the logics in this paragraph.

L. 220: "we observed"?

L. 220: "on these effectors". It is the beginning of a new paragraph. Please clarify to what "these effectors" refer to.

L. 235 "some of these structural T6SS genes adjacent to type III and type IV effector genes show homology." I don't understand the meaning of this sentence (and the following one on hcp). It is obvious that VgrG core components are part of a common family and are thus homologs sharing a common ancestry. Please clarify the message here. Do you mean that extra copies of VgrG (or hcp) come from intra-specific or intra-strain duplications?

Paragraph in lines 283-298: it could be noted that the core effectors are all part of the ancestral arsenal of the species. And yes, interestingly, all those in the MRCA are not core effectors, due to losses.

L. 291: when mentioning "stochastic mapping" as the method to infer the gene gains and losses, could the authors add a sentence about the principle of the method? To give an intuition to the reader about how it was done?

Figure 7b. I don't understand the meaning of the gray start "rhsP2 gain" on the rhsP2 gene tree itself as to me, the gains can only be mapped on the species tree.

L. 303: maybe cite the corresponding sup files?

L. 340 – to be consistent with the evolutionary scenario for the RhsP2 effector, it would have been interesting to check whether there could be traces in the genomic context supporting the multiple independent acquisitions. It is not evident to make the connection between this part and the one presented above and on Fig. 7, as the color code for strains harboring the effector are not consistent, and the strains selected for the displays are not the same. For instance, where are strains PA_151970, BWHPSA043, PPF-13 on Fig. 7d or 7e (or even 7b or 7c)?

Figure 8. Drawings are nicely done but the overall figure is very crowded. I am not sure whether all this material is necessary in main text, in particular the small graphs with the gain/loss rates? I am afraid the reader could get lost in the details here, whenever the drawings of the genomic loci do really help (along with the trees).

L. 446-447: "any two" to be replaced with "any pairs of"?

L. 556: please explain the PwCF abbreviation (or write it in full to avoid an unnecessary abbreviation as it is used only once).

L. 676-678. It is unclear how the use of the Secret6 database helped to delineate sub-groups of H4-T6SS. Please clarify.

L. 709. It is curious to note that the belonging to a T6SS sub-group is based on sequence identity and not gene clustering. Have the authors checked that all genes from a T6SS sub-group were all colocalized on the genome?

L. 788. Add "stochastic" before "mapping" to align with the typology in main text? In the following lines, please add which species tree is considered as none is mentioned.

L. 901-908. The testing of pairwise associations between effectors probably deserves a correction for multiple testing (for instance on the p-value threshold). Could the authors explain how it can be taken into account and apply the correction? This comment probably also applies to the association study between effectors and isolation sources depicted on lines 934-952.

Point-by-point response to the reviewer's comments

We would like to thank reviewer 3 for their time and effort in providing valuable feedback on the *Habich et al.* manuscript. Here, a summary of the main changes made to the manuscript and a point-by-point response to each comment below.

Main changes:

1. The evolutionary interpretation was strengthened by additional in-depth analyses of gene phylogenies and the genomic loci that encode each of the effector genes and the apparatus gene clusters. We found traces of mobile elements, detected recombination breakpoints, and determined events of vertical and horizontal gene transfer that explain the mechanisms driving the observed diversity.

The results of the additional analyses are presented in three new main figures (Fig. 3, 7, 8), four new sections of the main text, and over 100 additional display items in the supplementary files.

2. The methods section was expanded considerably with a detailed description of the performed analyses. As an indicator for the amount of information added, the methods section is now three times as long as it was before. Additional details were also added to the results section of the main text and the figure legends so that readers are able to follow better how individual analyses were performed.

Reviewer #3 (Remarks to the Author in black, our response in blue):

I thank the authors for this revised version of the manuscript. Unfortunately, to me the main criticisms made by reviewer #2 still hold as they were not sufficiently addressed. Here follow more details.

We thank the reviewer for their constructive assessment of the manuscript. More work was done to fully address each point, as described in more detail below.

Issues with evolutionary interpretation.

[1] Despite the authors' efforts to increase the discussion of the evolutionary aspect in relation to the observed diversity –in particular with the addition of the MRCA and effectors' dynamics part, I still have the feelings that this aspect is insufficiently covered while being indissociable of the work on the diversity of the effectors. Changing the title of the paper is not enough to alleviate the fact that it seems missing from the study. See also my comments below. In general, the mechanisms at the source of the origins and diversity of i) the different T6SS loci and ii) the different effectors could have been investigated in deeper details. On several occasions, potential mechanisms are invoked, but not further investigated. This includes in particular homologous recombination and lateral gene transfers. For instance, accompanying the gain/loss scenarios of T6SS effectors with phylogenetic analyses could have helped settling for vertical versus horizontal transfers (or duplications). Further, a study of the genomic locations and genomic context of the T6SS loci and effectors, could have helped to address the mechanisms of transfers. For instance, are they all located at the same loci? Are there traces of mobile genetic elements around? Etc... See here more specific comments:

Response:

We thank the reviewer for their constructive comments and suggestions.

Additional analyses on the effector genes (our response here and below to the related comments [1b] and [4]):

As suggested, we have performed much deeper analysis and provide additional evidence to support our manuscript. We have complemented the previously existing data on inferred effector gene gain and loss with new phylogenetic analyses of (i) the effector genes and (ii) the strains harbouring the effector genes. The co-phylogenies enabled us to conclude on vertical and horizontal gene transfer events (see Fig. 7 for one example on *rhsP2* in the main figure and Supplementary Data File 7 for all other effector genes). We observed two types of transfer: (i) lateral transfer of effector genes between strains of other species and *P. aeruginosa* strains that lead to the initial acquisition of an effector gene by *P. aeruginosa* strains and is followed by (ii) lateral and vertical transfer of effector genes between diverse strains of the species that lead to a patchy occurrence pattern amongst *P. aeruginosa* strains.

We have performed additional analysis to assess the genomic locations and genomic context as requested by reviewer 3. Our in-depth analysis found most effector genes encoded at the same loci adjacent to the same non-T6SS core gene, suggesting transfer by homologous recombination. One exception is *tle3* and *vgrG2b*, which we found at two loci among the analysed strains. Further, we detected recombination sites and present data on recombination breakpoints in vicinity to effector genes that suggests homology-facilitated illegitimate recombination and homologous recombination as the mechanisms of genetic gain of effector genes at four loci. We also found transposon-encoding genes in vicinity to effector genes in some strains, which are an additional hint to how effector genes might move as genomic islands.

The results of these additional analyses are shown in multiple new figures (Fig. 7,8). Three sections of new text have been added to the manuscript that are entitled “*Intraspecies inheritance of effector genes by vertical and lateral transfer among P. aeruginosa strains*”, “*Transfer of effector genes in T6SS genomic islands by homologous recombination and a potential role of transposons*”, and “*Frequent swapping of effector genes at four genomic loci by recombination*”.

Additional analyses on the T6SS loci (our response here and below to related comment [2]):

As suggested by the reviewer, we complemented our existing data on the predicted gain of the H4-T6SS with phylogenetic analyses. We found indications for four horizontal gene transfer events of this system from bacteria of other species to *P. aeruginosa* strains. Following an acquisition event from outside the species, our data suggests lateral and vertical transfer of the apparatus gene cluster among *P. aeruginosa* strains. We found traces of mobile genetic elements adjacent to genes encoding the H4-T6SS apparatus and not the other three apparatuses, which explains the scattered distribution of the H4-T6SS and not the other apparatus gene clusters among phylogenetically diverse strains of the species.

In regard to the H1-, H2-, and H3-T6SS apparatus gene cluster, we did not focus on the ancestry of these clusters as published work by Boyer *et al.* and Barret *et al.* has been performed on this showing that these clusters are ancient, were likely acquired independently by horizontal gene transfer, and did not arise by duplication within a *P. aeruginosa* strain (PMID: 18289922, 21474537). We acknowledge, that this was an oversight on our side and would likely be highly interesting to the non-T6SS reader. We have adjusted the text accordingly and performed additional analysis on our larger data set as requested by the reviewer. In support of this previous work, our data suggests that the three clusters were already present in the *P. aeruginosa* MRCA and a newly made comparison of the GC content of the apparatus genes with the average GC content of the genome (average GC%: 66.24) shows only minimal differences for the H2-T6SS (average GC%: 65.72), suggesting amelioration after ancient acquisition, and bigger differences for the H4-T6SS (average GC%: 63.24), suggesting recent acquisition. Unexpectedly we observed a higher GC content, compared to the rest of the genome, for the H1- and H3-T6SS suggesting more recent acquisition than the H2-T6SS and from a source other than the H4-T6SS. We feel this provided additional insight and aligns with the core message of reviewer 3 to strengthen the manuscript. We see no traces of mobile genetic elements next to the H1-, H2-, and H3-T6SS apparatus genes. We have also provided additional figures to demonstrate the loss events of

apparatus gene clusters among *P. aeruginosa* strains - which we are the first to report – that happens as part of large genomic deletions of varying size.

The results of these additional analyses are shown in new Fig. 3 and described in the new section in the text entitled “*Multiple independent gain and loss events of apparatus gene clusters during species diversification*”.

[1a] - Regarding the effectors' evolutionary history, reconstruction of individual gene phylogenies would have been needed to complement and support the gain/loss scenario. Gene trees should have been presented in a detailed manner. For instance, if only one acquisition event is inferred, it is expected that the gene history globally follows the species history after acquisition (and harbour the same topology). It does not seem enough to look for potential homologs in other bacteria as the authors did for some effectors. And even here, details are missing on how was led this search: search performed how? in which bacterial genomes? (see also point on Methods). As the authors say themselves, it “remains to be analysed in depth.”

Response:

In response to this comment, we have performed substantial new analysis by computing individual gene phylogenies anew for all effector genes and are now presented in a detail manner in new Fig. 7 for *rhsP2* and in 42 new figures in Supplementary Data File 7 for all other effector genes. As suggested, these phylogenies were used to assess the gain/loss scenario. Indeed, we observed strong indications for intraspecific inheritance following single transfer events in co-phylogenies of the gene tree and the species tree. Following such initial acquisition events, we observed and provide examples of inheritance of the effector genes among *P. aeruginosa* strains by vertical and horizontal transfer. Conclusions on vertical transfer were made from the gene tree matching the topology of the species tree. Conclusions on intraspecies lateral transfer were made from highly similar and sometimes identical effector genes that were detected in distantly related *P. aeruginosa* strains of different phylogroups.

We made additional new searches to identify homologs of the *P. aeruginosa* effector genes in the genomes of other species. As described in detail in the new methods section, we used megablast provided by the blastn suite of NCBI. The nucleotide sequences of *P. aeruginosa* effector genes were used as a query and the search was performed on the non-redundant standard databases of NCBI excluding sequences of *Pseudomonas aeruginosa* (taxid: 287). Among the hits provided by BLAST, genes with percent coverage above 80 and percentage identity above 70 were included in the tree of *P. aeruginosa* effector genes. Genomes harbouring these genes were included in the species tree of the co-phylogenies. By including these sequences, we observed that the effector genes of *P. aeruginosa* were more closely related to each other than to the closest relatives of other species, supporting our conclusions on initial acquisition events and subsequent vertical and horizontal transfer of the effector genes among strains of the species.

In sum, the results of these analyses are shown in new Fig. 7 and new Supplementary Data File 7 and described in a new section of the main text entitled “*Intraspecies inheritance of effector genes by vertical and lateral transfer among P. aeruginosa strains*”. We also substantially expand and detail the methods section as requested.

[1b] - More evolutionary analyses are discussed in the text (lines 277-280) for some of the effectors with a more scattered distribution. But only two were presented, in Figure 6D-E. What happened with the other effectors?

Response:

We thank the reviewer for raising this question. We have now added data for all effectors to the manuscript file, which resulted in over 40 additional figures. With a limit of 10 main display items and data on over 20 effectors, we decided to show one example on *rhsP2* in detail in new Fig. 7 as this provides an elegant example and present the data for all remaining effector genes in the supplementary files (Supplementary Data File 7).

[1c] - The authors used PhyloCorrelate in order to test for associations with traits (isolation source). However, PhyloCorrelate was designed to assess gene co-occurrence in a phylogenetic context. Could the authors elaborate on why they think that this method is well-suited to test the association between effector and traits?

Response:

Indeed, PhyloCorrelate was originally designed to study gene co-occurrences and became a widely used tool since its release in 2021. Importantly, the tool includes a phylogenetic correction, which takes the topology of the tree into account and avoids a co-occurrence being considered statistically significant purely based on the presence of two traits in the same lineage. The tool starts from a presence/absence matrix of the genes of interest across the genomes of interest. In our view, the methodology is equally valid whether the presence/absence matrix is built based on gene A (e.g. a genome harbouring effector gene *tne3* or not) and an isolation source B (e.g. the genome originating from a strain sampled from the environment or not) or gene A and gene B, as originally in mind by Tremblay et al. In both scenarios, the input data is binary and the tested traits have the potential to be linked. As Tremblay and colleagues suggest a functional association between co-occurring genes based on their use of the tool (e.g. genes of the same biosynthetic pathway), we suggest a functional association between an effector gene and the strain's environment that the effector protein is secreted into and acts on.

The methods section was expanded and now includes a justification for the use of this tool.

How and why H4-T6SS could be singular compared to H1/2/3.

[2] When reading the main text, it is still unclear what are the shared characteristics of the H4 T6SS and what makes them different from the other three loci. Especially when the authors name them H4a and H4b, whenever the phylogenetic tree seems to indicate there might be even more subclades. In Sup Fig 2, why not show the two loci H4a and H4b? It is actually shown later in Sup Fig 6 and it appears that the loci are almost identical. If the hypothesis of lateral gene transfers is invoked, why not pursue larger phylogenetic analyses to check the hypothesis of multiple acquisitions? (see also my general comment above).

Maybe these independent acquisitions could justify to give H4 different names for the different sub-clades.

Response:

We thank the reviewer for this comment.

We have improved the main text to make the differences clearer between the four different T6SS apparatus gene clusters and added additional analysis to further back up our statements. As suggested, we pursued larger phylogenetic analyses. Briefly, the H4-T6SS genes or components *tssB* or *tssM* are phylogenetically distinct (Fig. 1,3, Supplementary Fig. 1,9). These trees are built using *tssB*, *tssM* or a concatenated set of 12 genes to rigorously show that the H4-T6SS is a separate system from the H1/2/3.

Fig. 3 and Supplementary Fig. 9 and 10 show new or enhanced data that suggests four independent acquisitions of H4-T6SS apparatus gene clusters by *P. aeruginosa* strains from bacteria of other species and subsequent vertical and horizontal transfer of the clusters among *P. aeruginosa* strains. We no longer use the terminology 'H4a' and 'H4b' as there indeed may be more sub-clades and thank the reviewer for this comment.

We now show an alignment of apparatus gene clusters from all four acquisitions in Supplementary Fig. 9b. Importantly, the H4-T6SS gene clusters are very similar and share the same synteny. This synteny of the H4-T6SSs differs from the H1-, H2-, and H3-T6SS apparatus gene clusters, which also differ between each other in their synteny (Fig. 1b,c). In other words, apparatus gene clusters with the same synteny are given the same name and apparatus gene clusters with different synteny have different names (this also is corroborated by the phylogenetics discussed above and the H4-T6SS is never present in the loci of the H1-, H2- or H3-T6SS). Further, we identified traces of mobile genetic elements in vicinity of genes of the H4- and not H1-, H2-, H3-T6SS apparatus gene clusters, which could explain the movement of the H4-T6SS and its distinct occurrence pattern compared to the H1-, H2-, H3-T6SSs.

The results of these analyses are shown in new Fig. 3 (and new Supplementary Fig. 9, 10) and described in the new section "Multiple independent gain and loss events of apparatus gene clusters during species diversification". Existing Fig. 1 and existing text for this figure were adjusted to fully address this comment.

Database, dataset assembly and tree methods information are missing

There are still many missing pieces of information about the materials and methods used in this study. Here is a non-exhaustive list:

We thank the reviewer for bringing this to our attention. Major changes have been made to the methods section to describe the materials and methods in more detail.

It is still unclear what is the nature and quality of the genome datasets used.

[3a]- Line 107: please specify which dataset of *P. aeruginosa*. Also, from Methods and Sup mat, the genomes used seem not to be complete genomes. The authors should provide statistics on genome assembly state. Can they be considered of “high” quality? What is the level of completeness? Contamination? How does this affect T6SS annotation? None of this is discussed or detailed.

Response:

Details on the dataset of *P. aeruginosa* genomes and the rationale for the choice of this dataset were added to the text. A list with the accession numbers and metadata of the 1960 genomes used in this study is provided in Supplementary Table 2. Additional information on the dataset is provided in Supplementary Table 3,4.

The text now reads:
(Introduction, l. 61ff)

“In this study, we applied molecular population genetics to a dataset of ~2000 phylogenetically diverse *P. aeruginosa* strains (Supplementary Tables 2-4), focusing specifically on their T6SS-encoding genes. The publicly available genomes of the dataset are of high quality and represent the global *P. aeruginosa* population with strains from every continent, from humans through to animals and the environment, and exhibit remarkable phylogenetic diversity.”

(Results, l. 110ff (corresponding to l. 107 in previous manuscript version)

“To test for the distribution of the T6SS apparatus gene clusters within the species, we analysed their occurrence across strains. More specifically, we analysed a dataset of 1960 publicly available, high-quality *P. aeruginosa* genome sequences from strains of all continents and diverse sources (Supplementary Tables 2-4).”

Yes, out of a total of 1960 publicly available genomes that have been analysed, 239 genomes are complete. We consider the genomes in the dataset of high quality based on analyses with multiple tools (e.g. QUAST, BUSCO) and widely used thresholds for high quality genomes that are now described in detail in the methods section (see test passages below this paragraph). We are aware of potential effects of the genomes’ assembly state on the results and specifically controlled for this when performing the occurrence analysis by ensuring the genomic locus that is lacking an apparatus gene cluster or effector gene is located within a contig, which is now described in more detail in the methods section. Additionally, we performed the occurrence analysis of the apparatus gene clusters and the effector genes on the complete genomes only and recapitulated major findings made on the entire dataset (presence of H4-T6SS in few strains, occasional loss of H3-T6SS more often than occasional loss of H1-T6SS, same separation of effector genes into core and accessory effectors).

The text now reads:
(Methods, l. 635ff)

Accession of genomes and assessment of genome quality

We downloaded 1960 assembled *P. aeruginosa* genome sequences from NCBI using Entrez direct⁸⁴ on July 15th, 2022. We chose this dataset because the genomes cover

the phylogenetic diversity of the species, belong to strains that were isolated from every continent, and include strains of clinical and environmental origin (summarised in Supplementary Table 4). The genomes had previously been used in a large-scale bioinformatics study by Bortelho *et al.*, who performed an initial quality control and reported on a Mash distance between the genomes of 0.0001 to 0.05⁶⁸. Out of the 1960 genomes, 239 existed as complete genomes. We performed additional analyses to assess genome quality, completeness, and test for contamination. QUAST^{85,86} (version 5.2.0) was used to evaluate the quality of the genome assemblies. N50 values were between 111,311 and 7,564,383 with a mean of 1,624,041, which is over 1Mb and generally considered good. Genomes that consisted of more than one contig had L90 values between 2 and 86, which is below the threshold of 100 and an indication for little fragmentation⁸⁷. To assess the completeness of the assembled genomes, we tested for the presence of universal single-copy orthologs using BUSCO^{88,89} (version 5.7.0). The lineage dataset pseudomonadales_odb10 was used, which includes 782 universal single-copy orthologs for the order *Pseudomonadales*. Our genomes had BUSCO scores between 97.5% and 100%, which is considered very close to complete. To test for contamination, we analysed the genome's GC content using QUAST. The GC content of non-overlapping 100bp windows followed a Gaussian distribution (Supplementary Fig. 35), which is an indication for non-contaminated assemblies⁸⁵. In sum, we consider the 1960 genomes in our dataset of sufficient quality for our analysis, given the quality of assembly, level of completeness, and no sign of contamination. The accession codes and additional information on the genomes (source of isolation, sequence type (MLST), quality of assembly, level of completeness) are provided in Supplementary Table 2.”

(Methods, l. 705ff)

“Among the genomes with no T6SS hit, exemplary genomes were additionally inspected manually using the progressiveMAUVE algorithm in Geneious (version 2019.2.3) to ensure the genomic locus at which an apparatus gene cluster is expected is located within a contig and the absence of the gene cluster is not an artefact introduced by the assembly state of the genomes.”

(Methods, l. 728ff)

“Genomes with a hit above 80% identity for this locus without an effector gene were then considered negative for the respective effector and ensured that the absence of an effector gene at this locus is not an artefact introduced by the genomes' assembly state.”

We have performed additional validation using only the 239 complete genomes as stated here:

(Results, l. 117ff)

“These findings are recapitulated when analysing the complete genomes of the dataset only (n=239), excluding an artefact introduced by the genomes' assembly state (Supplementary Fig. 3).”

(Results, l. 260ff)

“Accordingly, we classify the *P. aeruginosa* effectors into six core and nine accessory effectors of the H1-T6SS, seven core and ten accessory effectors of the H2-T6SS, and two core and three accessory effectors of the H3-T6SS (Fig. 5c). This finding was recapitulated when analysing the occurrence of the effector genes only in the complete

genomes of the dataset with all three apparatus gene clusters (n=232) (Supplementary Fig. 12).”

[3b]- In link, line 195: what is the “global *P. aeruginosa* population”? Please explain dataset upon 1st mention in Results.

Response:

An explanation of the dataset was added at first mention and the wording adjusted. The dataset consists of a total of 1960 publicly available genomes of strains that were isolated from every continent, originate from diverse sources (humans, animals, environment), and are phylogenetically diverse (cover each *P. aeruginosa* phylogroup).

The text now reads:

(Introduction, l. 61ff)

“In this study, we applied molecular population genetics to a dataset of ~2000 phylogenetically diverse *P. aeruginosa* strains (Supplementary Tables 2-4), focusing specifically on their T6SS-encoding genes. The publicly available genomes of the dataset are of high quality and represent the global *P. aeruginosa* population with strains from every continent, from humans through to animals and the environment, and exhibit remarkable phylogenetic diversity.”

(Results, l. 110ff)

“To test for the distribution of the T6SS apparatus gene clusters within the species, we analysed their occurrence across strains. More specifically, we analysed a dataset of 1960 publicly available, high-quality *P. aeruginosa* genome sequences from strains of all continents and diverse sources (Supplementary Tables 2-4).”

More details are also required for the methods:

[3c]- Details are missing on how were the T6SS annotated. Blastn is invoked, but which parameters? And what is the “query coverage”? (line 445). What means “taking coverage into account”? Line 469: what “testing for the direct neighbourhood of genes” means?

Response:

Major changes were made to the methods sections on the occurrence analyses to provide sufficient detail and increase clarity on the use of blastn.

The text now reads:

(Methods, l. 683ff)

“Occurrence analysis of T6SS apparatus genes

A local blastn search was performed to measure the occurrence of the apparatus gene clusters directly in the dataset of assembled genomes without annotating the genomes. As queries, we used the individual genes of the three previously reported T6SS apparatus gene clusters of the reference strain PAO1 (H1-T6SS: PA0074-91, H2-T6SS: PA1656-71, H3-T6SS: PA2359-2373, PAO1 accession number: NC_002516)^{94,95}, and

the H4-T6SS apparatus gene cluster of the strains LYSZa7 (GCF_016584725.1) and 60.1 (GCF_900148005.1). As a database, we used the dataset of assembled genomes (Supplementary Table 2). The search was performed with nucleotide-nucleotide blast (version 2.10.1+) using default settings and adjusting `-max_target_seqs` to 5000. We chose the output format 10 with the following output parameters: `qseqid, qlen, qstart, qend, sacc, slen, sstart, send, qseq, sseq, length, nident, pident, bitscore, evalue`. Based on these parameters, we additionally calculated a nucleotide identity score of the blast hits to the query as described in detail by Rohwer *et al.*⁹⁶. In short, blast returned the percentage of identical matches ('pident') over the length of the alignment between the query and the subject, which did not take the total length of the query sequence into account. To account for the length of the query sequence, we calculated nucleotide identities like Rohwer *et al.* with the following equation using the parameters returned by the blastn analysis: $\text{nucleotide identity} = (\text{pident} * \text{length}) / (\text{qlen} + (\text{length} - (\text{qend} - \text{qstart} + 1)))$. Sequences with an identity score of 95% or higher were considered to be a hit. Genomes with hits for at least 12 apparatus genes were considered positive for the respective T6SS. All genomes in which we got blastn hits with an identity score below 95% or no blastn hit at all were additionally analysed with MacSyFinder⁹⁷ (version 2.1.1) (`--models TXSScan all`) to independently test for the presence of a T6SS apparatus gene cluster. Among the genomes with no T6SS hit, exemplary genomes were additionally inspected manually using the progressiveMAUVE algorithm in Geneious (version 2019.2.3) to ensure the genomic locus at which an apparatus gene cluster is expected is located within a contig and the absence of the gene cluster is not an artefact introduced by the assembly state of the genomes. To separate between genes of the H1-, H2-, H3- and H4-T6SS, we performed a reciprocal best hit analysis using blastn. The percentage occurrence of the apparatus genes was calculated based on the number of genomes with T6SS apparatus genes and the total number of genomes analysed.

Occurrence analysis of T6SS effector genes

To measure the occurrence of effector genes in the dataset of assembled genomes, we performed a search with blastn. Nucleotide sequences encoding for known *P. aeruginosa* T6SS effectors were extracted from annotated reference strains PAO1 (NC_002516), PA14 (NC_008463.1), PAK (NZ_CP020659.1), and from clinical strains PALA36 (CP110348) and PALA52 (CP110347), and used as queries (Supplementary Table 1). In case of *vgrG2b*, only nucleotides 2271 to 3060 were used as query because most of the protein encoded by this gene contributes to the structure of the T6SS and the enzymatically active domain with effector function is comparatively small. The blastn search and calculation of the nucleotide identity score were performed as described for the occurrence analysis of T6SS apparatus genes. Genomes with a hit above 80% nucleotide identity were considered positive for the respective effector gene. Among the genomes that were negative, individual genomes were inspected manually, the genomic locus identified at which an effector gene is located in other strains, the nucleotide sequence spanning one gene upstream and downstream of this locus extracted, and used as a query for a new blastn search. Genomes with a hit above 80% identity for this locus without an effector gene were then considered negative for the respective effector and ensured that the absence of an effector gene at this locus is not an artefact introduced by the genomes' assembly state. All genomes that had neither a positive nor a negative hit for an effector gene in the blastn search were further inspected manually with Geneious⁹³ (version 2019.2.3). The percentage occurrence was calculated based

on the number of genomes with a particular effector gene and the total number of genomes analysed. In the pan-genomic analysis, the established threshold of at least 95% occurrence for genes of the soft-core genome⁵² was applied to consider an effector gene a core effector or an accessory effector.”

[3d]- Line 476-477: “To separate effector proteins, the nucleotide sequences of effector genes that were encoded at the same location across genomes were translated and aligned using the Geneious alignment” – how was “the same location” defined?

Response:

We refer to T6SS genes being at the same genomic locus when they are at the same position in regard to the next adjacent non-T6SS gene of the core genome. The text of the results was adjusted accordingly and an entire section was added to the methods to explain our analysis of the genomic loci across strains in more detail.

The text now reads:

(Results, l. 141ff)

“Across the genomes, the H1-, H2-, and H3-T6SS apparatus gene clusters are encoded at the same respective genomic locus relative to the next adjacent non-T6SS gene of the core genome (Fig. 3a) [...]”

(Results, l. 335ff)

“Continuing with *rhsP2* as an example, we found *rhsP2* at the same locus (equivalent to PA14_43100 in the strain PA14) in-between the H2-T6SS apparatus gene cluster and the same non-T6SS gene of the core genome that is present across strains and across inferred acquisition events (Fig. 8a i).”

(Methods, l. 865ff)

“Analysis of genomic loci with T6SS genes across strains

To test whether T6SS apparatus or effector genes are located at the same genomic locus across acquisition events, we investigated the position of the gene(s) of interest relative to adjacent genes of the core genome. First, we took an exemplary genome that contained the gene(s) of interest and identified the closest non-T6SS genes of the core genome upstream and downstream. These genes of the core genome are expected to remain stable in the genome and serve as an anchor in our analysis. The information on the core genes was retrieved from the pan-genome analysis. Next, we used customised scripts to extract the nucleotide sequence spanning the first non-T6SS core gene upstream and downstream of the gene(s) of interest, and used this sequence as a query in a blastn search as described in the section on the occurrence analysis against all strains that encoded the gene(s) of interest. The percentage identity of the hit to the query was calculated as described in the section on the occurrence analysis. Hits with an identity score of > 95% were considered positive for the gene(s) of interest being adjacent to the same non-T6SS genes of the core genome. Strains with non-conclusive hits were analysed manually in Geneious⁹³ (version 2019.2.3).”

[3e]- Line 91: and on Figure 1C concatenation of which proteins?

Response:

The tree in Fig. 1d (former Fig. 1c), was built based on a concatenated alignment of the 12 apparatus genes *tssA*, *tssB*, *tssC*, *tssE*, *tssF*, *tssG*, *tssJ*, *tssK*, *tssL*, *tssM*, *clpV*, and *hcp* of the four apparatus gene clusters in strain 60.1. The details were added to the main text and figure legend.

The text now reads:

(Results, l. 95ff)

“To probe the phylogenetic distinction between the four clusters, we built a tree based on the concatenated alignments of 12 apparatus genes (*tssA*, *tssB*, *tssC*, *tssE*, *tssF*, *tssG*, *tssJ*, *tssK*, *tssL*, *tssM*, *clpV*, and *hcp*) of each of the four clusters from strain 60.1.”

(Main figure file, Figure legend, l. 16ff)

“Maximum-likelihood phylogenetic tree based on the concatenated alignments of 12 genes (*tssA*, *tssB*, *tssC*, *tssE*, *tssF*, *tssG*, *tssJ*, *tssK*, *tssL*, *tssM*, *clpV*, and *hcp*) encoding for structural proteins of the four T6SSs of strain 60.1.”

[3f]- Why use the GTR model for phylogenetic reconstruction, and not try to select the best evolutionary model (e.g. IQ-Tree does that automatically)?

Response:

In response to this comment, all trees were made anew using the function of IQ-Tree to automatically select the best evolutionary model. For each tree, information on the model chosen by IQ-Tree was added to the respective figure legend.

[3g]- How were the candidate donors found for H4-T6SS? Was it from a systematic search in all available complete genomes? Please explain the dataset used and general approach employed.

Response:

We performed a systematic search for proteins with homology to the *P. aeruginosa* H4-T6SS with blastp, using *P. aeruginosa* TssB4 as a query, and the non-redundant protein collection of NCBI as database. The species *Pseudomonas aeruginosa* (taxid 287) was excluded from the analysis. An entire section was added to the methods explaining our approach and the raw data with the results of the search is provided in Supplementary Data File 2.

The text now reads:

(Methods, l. 836ff)

“Search for homologs of *P. aeruginosa* TssB4 in other species

To identify homologs of *P. aeruginosa* TssB4 in other species, we used blastp provided by the blastp suite of NCBI. The protein sequences of TssB4 were used as query. One reason for this choice is that the nucleotide sequences did not reveal strong hits in a blastn search. *Pseudomonas aeruginosa* (taxid: 287) was excluded and the search was performed on the non-redundant protein sequences. BLAST hits with a query coverage

of above 95 percent (a measurement of BLAST to assess the length of the alignment between query and subject relative to the query length) and an identity of above 70 percent (a measurement of BLAST to assess the percentage identity along the alignment of query and subject) were considered homologs. Up to five top hits were chosen for further analysis. The genes encoding these TssB4 protein homologs and the genomes that harbour these genes were included in the co-phylogenies.”

[3h]- Why choose TssB to compute the phylogenies (Figure 1C), and not a conserved, longer protein? Also, please remind the proteins' role in T6SS upon 1st mention.

Response:

We used TssB because of its essential role in forming the outer sheath of the T6SS. Contraction of the outer sheath results in ejection of the inner tube outside of the bacterium and thereby facilitates the secretion of the effector proteins. TssB is very well understood, is considered a hallmark of the T6SS, conserved across type VI secretion systems, and multiple T6SS studies have used TssB for phylogenies (e.g. PMID: 25723169) so was our go to first choice for field-specific reasons.

To avoid any bias that might be introduced based on the choice of TssB, we also made phylogenies based on TssM, which is an essential component of the membrane complex that anchors the T6SS in the cell membrane. TssM is roughly six times longer than TssB. We observed a very similar topology of the tree than for TssB (Supplementary Fig. 1).

An addition was made to the main text to explain the rationale for the choice of TssB and added details on its role in the T6SS.

The text now reads:

(Results, l. 100ff)

“Further confirming this result, phylogenetic trees of two apparatus genes *tssB* and *tssM* that encode proteins with essential T6SS functions as components of the outer sheath (TssB) and membrane complex (TssM) show clear cladding into four branches - one for each of the H1- to H4-T6SS (Supplementary Fig. 1).”

[3i]- Line 179: one cannot measure such low sequence identity, this makes no sense as sequences are too divergent to be aligned. Please clarify.

Response:

The text was changed and now refers to the divergence of the proteins.

It now reads:

(Results, l. 398f)

“TspE1b and TspE1c are too divergent to be properly aligned (Supplementary Fig. 19a) and expected to differ in their pyocin-like effector activity²⁵.”

[3j]- Line 223: the general principles of the approach used to infer ancestral states is not

mentioned in Results, neither are provided the details of which model among the four tested was finally used to produce the results.

Response:

In response to this comment, additions were made to the results to explain the principle at first mention. The methods section was expanded and now describes that all four models were used and the weight given to each model was determined by its relative support. Information on model weights is provided in Supplementary Table 7 and Supplementary Data File 3. In addition, results of each ancestral mapping are now displayed in Supplementary Data File 1,4,9,10 (files open correctly in Preview (MacOS) or Microsoft Edge (MacOS and Windows)).

The text now reads:

(Results, l. 137ff)

“To investigate the source of diversity in the intraspecific occurrence of apparatus gene clusters, we used stochastic mapping. This method aims to reconstruct the evolutionary history of a gene on a phylogenetic tree of the species.”

(Methods, l. 787ff)

“The R package `phytools`^{105,106} was used for the mapping as described by Revell (2024)¹⁰⁶. More specifically, we used effector presence and absence as a binary trait and fitted four character transition models (ER: equal rates model, ARD: all-rates-different model, and two different irreversible trait evolution models) using the `fitMk()` function. The fitted models were then compared and their Akaike Information Criterion (AIC) was calculated using the generic `anova()` function. **Stochastic character maps were sampled with the `simmap()` function (`nsim=1000`) using all four fitted models weighted by their relative frequency equal to the evidence supporting each model (Akaike model weights)¹⁰⁶.** We then calculated the posterior probabilities for all effector genes at each ancestral node with the `summary()` function.”

Questionable usefulness of the statistical analysis of combinations of effectors

[4a] - I still do not really see the point of the effectors' combination analysis, especially when means for the diversification of these combinations are not discussed.

Response:

We thank the reviewer for making us aware that we were not sufficiently clear in why we analysed the combination of effectors in effector sets and what the implications of the diverse effector sets are and made the following changes:

1. The text was expanded and now lays out our three reasons for studying the effector sets.

It now reads:

(Results, l. 430ff)

“Next, we investigated the co-occurrence of effector genes that arose in the genomes by inheritance from the species' MRCA and subsequent gain, loss, and exchange of individual effector genes. The combination of effector genes associated with one T6SS

apparatus in a given strain is referred to as an 'effector set'. We had three reasons for this analysis: First, multiple different effector proteins are secreted simultaneously with one shot of the secretion apparatus they are associated with and there might be steric hindrance or synergy between effectors that could affect the co-occurrence of effector genes in a genome positively or negatively over time. Second, the T6SS apparatuses differ in their regulation and the strategic use of the individual apparatus might be reflected in the diversity of the effector genes that are associated with the apparatus. Third, most effectors play a role in bacteria-bacteria killing and in particular intraspecies killing, thus the effectors that two strains attack each other with, and their presence in the genome, serves as a proxy for the ability of two strains to co-exist or kill one another."

2. Data on the implications of the diverse effector sets was added to new Fig. 10. Additionally, implications of the diverse effector sets are now described in the main text and discussion.

It now reads:

(Results, l. 533ff)

"Fifth, we tested effector occurrence according to the strategic use of the T6SS in interbacterial competition (Fig. 10d). The H1-T6SS is considered a defensive weapon because it is inactive unless the bacterium is being attacked and the H1-T6SS is being activated in response to fight off the attacker^{63,64}. In contrast the H2-T6SS is considered offensive because it is activated at high cell density and does not depend on an external cue to engage in killing other bacteria⁶⁵. Both systems have similar numbers of core and accessory effectors with some in the MRCA and others not. Interestingly, a smaller number of diverse effector sets was observed for defensive use by the H1-T6SS (Fig. 9a iv) whilst those for offensive use by the H2-T6SS had a much higher number of diverse effector sets driven by the larger number of accessory effectors that were encoded at varying genomic loci and not as mutually exclusive effector genes at one locus (Fig. 9a viii)."

(Discussion, l. 616ff)

"With a limit to the number of effectors that can be secreted with one shot of the T6SS, there may not be an advantage from simply accumulating effector genes. The exchange of effector genes directly embedded in a regulatory network and in a manner that guarantees secretion with existing structural proteins encoded in a specific T6SS genomic island seems a very elegant mechanism that could allow rapid adaptation to the ebb and flow of diverse bacterial competitors in the immediate environment. Additionally, the acquisition of a new rare effector encoding gene could allow them to displace their ancestors due to the negative frequency-dependent selective advantage."

(Discussion, l. 626ff)

"Finally, we found an extremely large biodiversity in T6SS effector sets in a natural population of *P. aeruginosa*. Indeed, laboratory strains neither encode all T6SSs of the species nor do they represent the most prevalent combined effector set."

3. The figure on the effector sets (now Fig. 9) was moved to an earlier position in the manuscript so that the motivation for the analysis and the implications of the diverse effector sets better match the flow of the manuscript.

[4b] - The occurrence analysis does not take phylogenetic inertia into account (as the authors tell themselves), whenever it is responsible of most of the co-occurrences observed – as further questioned by the very few positive associations found. I question the relevance of keeping it in the text.

Response:

In response to the comment, this occurrence analysis was removed from the main text.

Some concepts are clumsily introduced and convoluted.

[5] - Lines 157-160. I did not understand the concept of “mutually exclusive effectors” when reading the manuscript at first. Could the authors please clarify? I only could understand because I read the authors’ response to reviewer #2. Also, no evolutionary mechanism is discussed here. It kind of falls flat, which is a pity as the sources of this diversity would be very interesting to unravel (see my above comment).

Response:

We share the reviewer’s interest in the diversity of these effectors, performed additional analyses, and clarified the text.

For clarification of the concept, a graphical depiction was added to Fig. 8e iv showing closely related strains in which an effector gene was gained by replacement of the previously existing effector gene. We observed such a gain by replacement at four genomic loci.

To address the mechanism, we tested for recombination within these genes using the established tool RDP5 and identified multiple recombination sites. For example, a strong recombination breakpoint was observed upstream of *tspE1b*, suggesting homology-facilitated illegitimate recombination and the acquisition of *tspE1b* at this site by replacement of *tspE1c* (Fig. 8e v).

As a consequence, strains either encode *tspE1b* or *tspE1c* at this genomic locus and never both simultaneously, which results in them being mutually exclusive in their occurrence pattern across strains of the species.

In sum, the new data is displayed in new Fig. 8e,f and new Supplementary Fig. 22,23,24. An entirely new section of text was added to highlight these findings in the result section, entitled “Frequent swapping of effector genes at four genomic loci by recombination”. The findings are further built upon in the discussion:

The text in the discussion was expanded and now reads:

(Discussion, l. 613ff)

“What is particularly striking to us are the T6SS genomic islands with *vgrG* genes, which likely arose by duplication as observed for the *vgrG1abc* or *vgrG2a/2b* islands, and the subsequent much more frequent swapping of effector genes within these T6SS islands by homology-facilitated illegitimate recombination. With a limit to the number of effectors

that can be secreted with one shot of the T6SS, there may not be an advantage from simply accumulating effector genes. The exchange of effector genes directly embedded in a regulatory network and in a manner that guarantees secretion with existing structural proteins encoded in a specific T6SS genomic island seems a very elegant mechanism that could allow rapid adaptation to the ebb and flow of diverse bacterial competitors in the immediate environment. Additionally, the acquisition of a new rare effector encoding gene could allow them to displace their ancestors due to the negative frequency-dependent selective advantage.”

Other comments

[6a] - Why not explicitly state in which strain(s) the H4 cluster was first discovered? From the text, it seems that it has been missed/overlooked from many studies of *Pseudomonas aeruginosa*'s genomes. Yet it is found in only a few *P. aeruginosa* strains – which explains why it has been so far overlooked.

Response:

The *tssM4* gene of the H4 cluster had first been discovered in the strain LYSZa7. As suggested, the name of this strain was now added to the text.

The sentence now reads:

(Results, l.90ff)

“This cluster includes genes for 12 of the 13 required structural T6SS components (one of them being *tssM4* recently mentioned in the strain LYSZa7 by Ren *et al.*²⁷).”

[6b] - Figure 1: coloring gene families according to family of homologs would help to visualize genomic organization commonalities/differences between the loci.

Response:

As suggested, colours were added to the genes in Fig. 1b,c and chosen to best highlight the differences in synteny between the clusters.

[6c] - What are cargo and specialized effectors? Please define.

Response:

The terms cargo effector and specialized effectors have been introduced by others in the T6SS field multiple years ago and are used by us for consistency with the existing literature. The definitions are:

Specialized effectors are proteins that have a functional role in assembly of a fully functional secretion system and additionally have effector activity. These proteins often consist of multiple domains, one domain that forms an essential structural component of the secretion system and a second domain that mediates effector activity. An example is VgrG2b that forms

parts of the tip of the secretion system and has a C-terminal extension that mediates enzymatic activity.

Cargo effectors are proteins with effector activity that are secreted by the T6SS without being required for a functional secretion apparatus. An example is Tse1 that is secreted by the T6SS and toxic to other bacteria it is delivered to. Tse1 itself has no structural role in the secretion apparatus and deletion of *tse1* had no effect on the functionality of the T6SS (PMID: 20114026). For secretion Tse1 is loaded into a Hcp ring within the tube of Hcp proteins which is propelled out of the system upon contraction of the TssB/TssC sheath and into a competing bacterial cell.

Definitions of the terms have been introduced to the text and graphic depictions of cargo and specialized effectors were added to new Fig. 10.

The text now reads:

(Results, l. 520ff)

“We found that cargo effectors, which are defined as effector proteins with a non-covalent bond to structural T6SS components^{11,62}, were the most common amongst the core effectors.”

Point-by-point response to the reviewer's comments

We would like to thank reviewer 3 for their time and effort in providing valuable feedback on the *Habich et al.* manuscript. Below our responses to each comment.

Reviewer #3 (Remarks to the Author in black, our responses in blue):

I would like to thank the authors for the considerable amount of work and efforts they put in this round of revision. I am happy with the replies to my comments and my concerns have been addressed. I believe this article now provides a lot of insights on the mechanisms of T6SS' and effector sets' diversification in *Pseudomonas aeruginosa*. The study addresses understudied, long-standing issues in the biology of secretion, in particular that of the T6SS. The unraveled diversity of the evolutionary mechanisms involved here should be very interesting to anyone interested in bacterial evolution. Moreover, the methods are now provided with a level of details sufficient for the reproducibility of the results and the discussion of the possible limits of the methodology. I still have some minor comments, as they are many new elements to account for.

We thank the reviewer for their very positive assessment.

[1] In general, support values for phylogenetic trees are missing, clouding the interpretation of the significance of branchings. These should be added. It could be just an indication of the branches with a support e.g. >90% or 95%. This is particularly missing for Fig 3e.

As suggested, support values were added to phylogenetic trees in Figure 3, 6, 7, 8.

[2] The statistical tests results should be provided whenever they are mentioned (e.g. name or type of test, and p-value).

The details on the statistical tests were added and the text now reads:

I. 447ff

"No statistically significant co-occurrence (**Hypergeometric test, false discovery rate (FDR) = 0.05, $P < 0.05$**) was observed between any pairs of accessory effectors encoded in various combinations at the loci equivalent to PA0093 and PA0099 (Supplementary Fig. 25, Supplementary Data 25)."

I. 464ff

"Testing for the co-occurrence between pairs of accessory effectors that are encoded at different genomic loci revealed one statistically significant association (**Hypergeometric test, FDR = 0.05, $P < 0.05$**) between the effector genes *pldA* and *tseV* (Supplementary Fig. 25, Supplementary Data 25)."

I. 561ff

"Remarkably, amongst the strains of animal, human, and environmental origin, we found *tIe3* to be associated with animal origin (**Hypergeometric test, FDR = 0.05, $P < 0.05$**) (Fig. 10e i-iii, Supplementary Data 27)."

[3] Fig.1. Please indicate what the colors correspond to? Or a shared color? Would it be possible to add on the figure the strains with 2 loci H4-T6SS?

As suggested, a legend was added to indicate the structural parts of the T6SS apparatus that the colours correspond to.

We prefer not to add additional strains with the H4-T6SS for consistency with the display of the H1-, H2-, and H3-T6SSs. There are no strains with 2 loci for the H4-T6SS, rather two different locations of insertion when comparing different genomes, this is shown in Figure 3a and detailed in Supplementary Figure 7. An alignment with the H4-T6SS of multiple strains is already provided in Supplementary Figure 8b.

[4] Fig. 3. The font size is small.

We understand that this applied in particular to the labelling of the phylogenetic tree in Fig. 3f and increased the font size accordingly.

[5] L. 212-215. I don't understand here, as first the number of effectors of strain PAO1 is mentioned, and then it is said that "Effectors of H4-T6SS were omitted here". But is not PAO1 missing the H4-T6SS locus? As shown on Fig 2d? Please clarify the logics in this paragraph.

Indeed, PAO1 is missing the H4-T6SS locus. The paragraph was clarified and the sentence of question was moved to a different section. The text now reads:

I. 213ff

"Each effector is associated with one of the three T6SS-apparatuses it is secreted by. ~~Effectors of the H4-T6SS were omitted here as none have been characterised to date.~~ In three instances in PAO1, individual effector proteins have a dual function as a structural component and an effector (VgrG2b, Tse6 and Tse7)."

I. 242ff

"To test for the intraspecific distribution of effector genes, we analysed their occurrence in our dataset of diverse *P. aeruginosa* genomes from around the world. To avoid a bias because of the lack of a T6SS apparatus gene cluster in some strains, we analysed only those genomes of the dataset with H1-, H2- and H3-T6SS apparatus genes (n=1912). **Effectors of the H4-T6SS were omitted here as none have been characterised to date.** We found some effector genes widely distributed and some others with patchy distribution (Fig. 5a, Supplementary Data 10-12)."

[6] L. 220: "we observed"?

The sentence was corrected accordingly and now reads:

I. 219f

"Focusing our attention on the genomic loci of effector genes, we **observed** their distribution across the genome of *P. aeruginosa* as depicted for PAO1 (Fig. 4b)."

[7] L. 220: "on these effectors". It is the beginning of a new paragraph. Please clarify to what "these effectors" refer to.

The sentence was modified to increase clarity and now reads:

I. 219ff

“Focusing our attention on **the genomic loci** of effector genes, we observed their distribution across the genome of *P. aeruginosa* as depicted for PAO1 (Fig. 4b).”

[8] L. 235 “some of these structural T6SS genes adjacent to type III and type IV effector genes show homology.” I don't understand the meaning of this sentence (and the following one on *hcp*). It is obvious that *VgrG* core components are part of a common family and are thus homologs sharing a common ancestry. Please clarify the message here. Do you mean that extra copies of *VgrG* (or *hcp*) come from intra-specific or intra-strain duplications?

Our main message of this paragraph is the genomic organization of effector genes. This comment made us realize that the sentences on *vgrG* and *hcp* might be obvious and distracting, we therefore moved the corresponding panels to the supplement. The homology between *vgrG* genes was further analysed by sequence alignments, which were added to Supplementary Fig. 11 and suggest intra-specific duplication events that gave rise to *vgrG2a*, *vgrG2b* and *vgrG4a*, *vgrG4b*.

The main text now reads:

I. 230ff

“These additional T6SS structural genes may be *hcp*, *vgrG* or PAAR that encode T6SS structural components required for export of the respective effector and are secreted themselves (Fig. 4a,c,d, Supplementary Fig. 11). In rare cases, type III or IV effector genes are additionally encoded next to apparatus gene clusters (e.g. *vgrG1a*, *tse6*, and *tsi6* next to the H1-T6SS) (Fig. 4b). Considering the various loci of effector genes within a single genome and their presence either as isolated genes or in the neighbourhood of other T6SS genes made us wonder about the occurrence of effector genes in phylogenetically diverse strains of the species.”

[9] Paragraph in lines 283-298: it could be noted that the core effectors are all part of the ancestral arsenal of the species. And yes, interestingly, all those in the MRCA are not core effectors, due to losses.

As suggested, additional sentences were added and the text now reads:

I. 273ff (added sentences in bold)

“The MRCA harboured eight effector genes of the H1-T6SS, eight effector genes of the H2-T6SS, and three effector genes of the H3-T6SS (Fig. 6b, Supplementary Fig. 14, Supplementary Data 16, 17). **The core effector genes were all present in the MRCA. However, not all effector genes of the MRCA are core effectors in our dataset of diverse strains, which made us wonder about the loss of effector genes in the species' evolutionary history.**”

[10] L. 291: when mentioning “stochastic mapping” as the method to infer the gene gains and losses, could the authors add a sentence about the principle of the method? To give an intuition to the reader about how it was done?

Following this suggestion, additional sentences on the principle of stochastic mapping were added:

I. 137ff (addition in bold)

“This method aims to reconstruct the evolutionary history of a gene on a phylogenetic tree of the species **by determining the likelihood for the gene’s presence at each node of the tree.**”

I. 272f

“This method determines the likelihood for the presence or absence of an effector gene at each node along the branches of the tree.”

[11] Figure 7b. I don’t understand the meaning of the gray star “*rhsP2* gain” on the *rhsP2* gene tree itself as to me, the gains can only be mapped on the species tree.

This is a good point, the grey star was removed to increase clarity.

[12] L. 303: maybe cite the corresponding sup files?

A reference to the corresponding Supplementary Data Files was added. The sentence now reads:

I. 301ff (addition in bold)

For space reasons, we focus on *rhsP2* as an example in the main text and provide data for the remaining effector genes in supplementary files (**Supplementary Data 18-21**).

[13] L. 340 – to be consistent with the evolutionary scenario for the *RhsP2* effector, it would have been interesting to check whether there could be traces in the genomic context supporting the multiple independent acquisitions. It is not evident to make the connection between this part and the one presented above and on Fig. 7, as the color code for strains harboring the effector are not consistent, and the strains selected for the displays are not the same. For instance, where are strains PA_151970, BWHPSA043, PPF-13 on Fig. 7d or 7e (or even 7b or 7c)?

The Fig. 7c was adjusted to enable a direct comparison between the highlighted acquisition events in Fig. 7 and Fig. 8 and smooth the transition between sections and figures.

[14] Figure 8. Drawings are nicely done but the overall figure is very crowded. I am not sure whether all this material is necessary in main text, in particular the small graphs with the gain/loss rates? I am afraid the reader could get lost in the details here, whenever the drawings of the genomic loci do really help (along with the trees).

We understand that the figure is crowded despite our best efforts. Our preferred option would have been to split the figure into two, which is not possible due to the limited number of main display items. As mentioned by the reviewer, the individual graphs do help grasping the observed complexity of the gain, loss, and exchange of effector genes and we therefore would like to keep the figure in its current state as its important to the core message of the story that there is more than one example effectively conveyed.

[15] L. 446-447: “any two” to be replaced with “any pairs of”?

The sentence was modified as suggested. It now reads:

I. 447ff

“No statistically significant co-occurrence (Hypergeometric test, false discovery rate (FDR) = 0.05, $P < 0.05$) was observed between **any pairs of** accessory effectors encoded in various combinations at the loci equivalent to PA0093 and PA0099 (Supplementary Fig. 25, Supplementary Data 25).”

[16] L. 556: please explain the PwCF abbreviation (or write it in full to avoid an unnecessary abbreviation as it is used only once).

The abbreviation ‘PwCF’ is no longer used in the text.

[17] L. 676-678. It is unclear how the use of the Secret6 database helped to delineate subgroups of H4-T6SS. Please clarify.

We realized that the original sentence was misleading. The subgrouping here refers to a classification system in the T6SS field across T6SSs of diverse bacterial species and not to subgroups within the H4-T6SSs. The sentence was adjusted and now reads:

I. 685ff

“To classify the H4 apparatus gene cluster according to an established classification system of the T6SS field that is based on very diverse T6SSs from a wide range of bacterial species, we used the SecReT6 database⁹⁶ and the amino acid sequence of TssB4 of strain 60.1 as a query.”

[18] L. 709. It is curious to note that the belonging to a T6SS sub-group is based on sequence identity and not gene clustering. Have the authors checked that all genes from a T6SS sub-group were all colocalized on the genome?

Indeed, the synteny of genes within the H1-, H2-, H3-, or H4-T6SSs is conserved across strains of the dataset. We did check and added this information to the manuscript:

I. 722ff

“The co-localization of apparatus genes in a cluster was checked separately by computing multiple sequence alignments of the entire gene cluster and confirmed (H1-T6SS: n=1605, H2-T6SS: n=1949, H3-T6SS: n=1883, H4-T6SS: n=19; all genomes with apparatus genes on one contig).

[19] L. 788. Add “stochastic” before “mapping” to align with the typology in main text? In the following lines, please add which species tree is considered as none is mentioned.

The text in the paragraph was adjusted to uniformly refer to “stochastic mapping” and to provide sufficient detail on the species tree:

I. 792ff

“To reconstruct the presence or absence of effector genes in ancestral strains of the species, we performed stochastic character mapping. Therefore, the R package phytools^{110,111} was used as described by Revell¹¹¹. Each effector gene was mapped onto each node of the species tree. This phylogenetic tree was built as described in the separate section of the methods based on 1912 *P. aeruginosa* genomes (all genomes in the dataset with all three apparatus genes clusters of the H1-, and H2-, and H3-T6SS) and an outgroup of five genomes from the closely related species *P. paraaeruginosa* and seven genomes of distinct *Pseudomonas*

species (*P. delhiensis*, *P. humi*, *P. jinjuensi*, *P. knackmussii*, *P. multiresinivorans*, *P. nitroreducens*, *P. panipatensis*) that had also been used as an outgroup in an existing study⁵³ (accession codes of all genomes used are listed in Supplementary Data 15).”

[20] L. 901-908. The testing of pairwise associations between effectors probably deserves a correction for multiple testing (for instance on the p-value threshold). Could the authors explain how it can be taken into account and apply the correction? This comment probably also applies to the association study between effectors and isolation sources depicted on lines 934-952.

As suggested, the correction for multiple testing was applied to both analyses and the text and figures were adjusted accordingly. The text now reads:

I. 447ff

“No statistically significant co-occurrence (**Hypergeometric test, false discovery rate (FDR) = 0.05, $P < 0.05$**) was observed between any pairs of accessory effectors encoded in various combinations at the loci equivalent to PA0093 and PA0099 (Supplementary Fig. 25, Supplementary Data 25).”

I. 464ff

“Testing for the co-occurrence between pairs of accessory effectors that are encoded at different genomic loci revealed one statistically significant association (**Hypergeometric test, FDR = 0.05, $P < 0.05$**) between the effector genes *pIdA* and *tseV* (Supplementary Fig. 25, Supplementary Data 25).”

I. 561ff

“Remarkably, amongst the strains of animal, human, and environmental origin, we found *tIe3* to be associated with animal origin (**Hypergeometric test, FDR = 0.05, $P < 0.05$**) (Fig. 10e i-iii, Supplementary Data 27).”